# DeWave: Discrete EEG Waves Encoding for Brain Dynamics to Text Translation

**Yiqun Duan**[1][*], **Jinzhao Zhou**[1], **Zhen Wang**[2], **Yu-Kai Wang**[1], **Chin-Teng Lin**[1][†]

[1]GrapheneX-UTS HAI Centre, Australian Artificial Intelligence Institute,
Faculty of Engineering and Information Technology
University of Technology Sydney, Ultimo, NSW 2007
[2]School of Computer Science, The University of Sydney, Camperdown NSW 2050
{yiqun.duan, jinzhao.zhou}@student.uts.edu.au,zwan4121@uni.sydney.edu.au
yukai.wang@uts.edu.au, chin-teng.Lin@uts.edu.au

## Abstract

The translation of brain dynamics into natural language is pivotal for brain-computer interfaces (BCIs). With the swift advancement of large language models, such as ChatGPT, the need to bridge the gap between the brain and languages becomes increasingly pressing. Current methods, however, require eye-tracking fixations or event markers to segment brain dynamics into word-level features, which can restrict the practical application of these systems. To tackle these issues, we introduce a novel framework, DeWave, that integrates discrete encoding sequences into open-vocabulary EEG-to-text translation tasks. DeWave uses a quantized variational encoder to derive discrete codex encoding and align it with pre-trained language models. This discrete codex representation brings forth two advantages: 1) it realizes translation on raw waves without marker by introducing text-EEG contrastive alignment training, and 2) it alleviates the interference caused by individual differences in EEG waves through an invariant discrete codex with or without markers. Our model surpasses the previous baseline (40.1 and 31.7) by 3.06% and 6.34%, respectively, achieving 41.35 BLEU-1 and 33.71 Rouge-F on the ZuCo Dataset. This work is the first to facilitate the translation of entire EEG signal periods without word-level order markers (e.g., eye fixations), scoring 20.5 BLEU-1 and 29.5 Rouge-1 on the ZuCo Dataset.

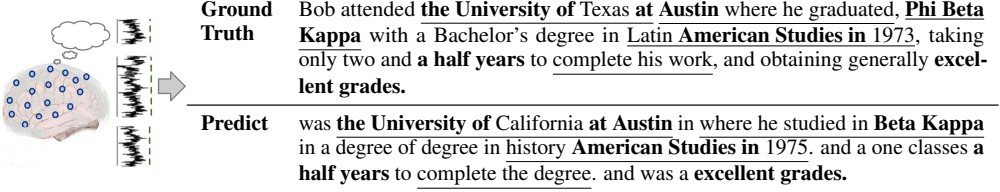

| | | |
|---|---|---|
| **Ground Truth** | Bob attended **the University of** Texas **at Austin** where he graduated, **Phi Beta Kappa** with a Bachelor's degree in Latin **American Studies in** 1973, taking only two and **a half years** to complete his work, and obtaining generally **excellent grades.** |
| **Predict** | was **the University of** California **at Austin** in where he studied in **Beta Kappa** in a degree of degree in history **American Studies in** 1975. and a one classes **a half years** to complete the degree. and was a **excellent grades.** |

Figure 1: Overall illustration of translating EEG waves into text [2] through quantised encoding.

## 1 Introduction

Decoding brain states into comprehensible representations has long been a focal point of research [20, 38, 8, 54]. Electroencephalogram (EEG) signals are particularly favored by researchers due to their non-invasive nature and ease of recording [31, 46]. Traditional EEG decoding techniques

---

[*]is the first author, † is the corresponding author

[2]This visualization still keeps teacher-forcing evaluation for a fair comparison with previous methods.

37th Conference on Neural Information Processing Systems (NeurIPS 2023).

largely focus on classifying brain states into restricted categories like Motor Imaginary (MI) [37, 7], Emotion [19, 47], Robotic Control [44, 55], and Gaming [28, 24]. However, these labels, bound to specific tasks, are insufficient for broad-based brain-computer communication. Consequently, there has been a surge of interest in brain-to-text (speech) translation in recent years. As the current trend leans towards large language models (LLM) [4, 12, 30] showcasing increasingly generalized intelligence capabilities, it becomes crucial to delve into ways of bridging the gap between brain signals and natural language representation. However, this area remains under-explored.

The early work in brain-to-text translation [14, 1, 26, 41] relied on external event markers like handwriting or eye-tracking fixations to segment whole brain signals into fragmentary features. This methodology treated the task as word-level classification on a small, closed vocabulary set, with each time step analyzed individually. Both invasive [14, 42] (ECoG) and non-invasive [33] (EEG) brain signals have been used in these approaches. Notably, utilizing handwriting as event markers on invasive signals has led researchers [49] to achieve state-of-the-art (SOTA) recognition accuracy on closed-set character-level recognition. Wang [48] expanded the vocabulary size substantially and demonstrated the feasibility of open-vocabulary brain-to-text translation by employing pre-trained language models with word-level EEG features. However, limitations persist. The order of event markers, particularly eye-tracking fixations used to segment EEG waves into word-level features, may not match the natural word order in language [3]. Moreover, current methods do not have the capacity for direct text translation.

In this paper, we present Discrete EEG Waves Encoding for Brain Dynamics to Text Translation (DeWave), a pioneering framework depicted in Figure 2. DeWave uses a vector quantized variational encoder to transform EEG waves into a discrete codex, linking EEG waves to tokens based on their proximity to codex book entries. This method offers two key advantages: it addresses significant distribution variances in EEG waves across individuals [29, 40, 9], and rectifies order mismatches between raw wave sequences and text without eye-tracking markers. Our raw waves encoder is guided by both self-reconstruction and a contrastive supervision alignment between text embeddings and vectorized raw waves. To navigate the challenges of training with limited parallel data, DeWave leverages large-scale pre-trained language models [6, 36, 3], specifically employing BART [21], which combines BERT's bidirectional context with GPT's left-to-right decoder. Notably, our discrete codex aligns more closely with actual language tokens than continuous EEG features, serving as an interpretable bridge between EEG input and the language model.

Experiments employ non-invasive EEG signals and data from the ZuCo dataset [16], a large-scale public resource that records eye-tracking and EEG during natural reading tasks. Notably, DeWave can be generalized for both word-level EEG features and raw EEG wave translation. With a robust codex representation, this work pioneers in translating the entire time period of EEG signals without the need for word-level order markers such as eye fixations. We assess the performance using standard translation metrics [34, 25]. With word-level EEG features, DeWave attains 42.8 BLEU-1 and 34.9 Rouge-1, surpassing the previous baseline (40.1 and 31.7) by 6.73% and 10.09% respectively on the ZuCo Dataset. For raw EEG waves without event markers, DeWave achieves 20.5 BLEU-1 and 29.5 Rouge-1. This work's contributions can be summarized in three main points.

- This paper introduces discrete codex encoding to EEG waves and proposes a new framework, DeWave, for open vocabulary EEG-to-Text translation.

- By utilizing discrete codex, DeWave is the first work to realize the raw EEG wave-to-text translation, where a self-supervised wave encoding model and contrastive learning-based EEG-to-text alignment are introduced to improve the coding ability.

- Experimental results suggest the DeWave reaches SOTA performance on EEG translation, where it achieves 41.35 BLUE-1 and 33.71 Rouge-1, which outperforms the previous baselines by 3.06% and 6.34% respectively.

## 2 Related Works

The key to decoding natural language from EEG signals is good representations. Existing work for EEG-to-text representation can be categorized into hand-crafted representations and deep-learning

---

[3]For example, the ZuCo Dataset collects data by simultaneously recording eye-tracking fixation and brain waves during reading tasks. However, the order of eye-fixations and spoken words may not always coincide.

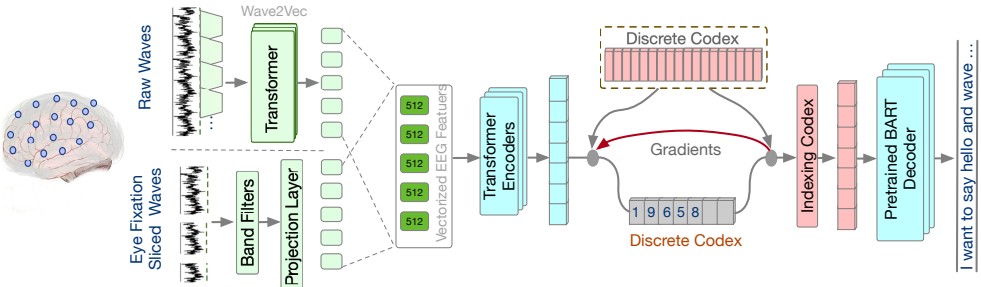

Figure 2: The DeWave model structure involves vectorizing either word-level EEG features or raw EEG waves into embeddings (Section 3.3). The vectorized features are then encoded into a latent variable $\mathbf{z}_c(\mathcal{X})$, which is converted into a discrete latent $\mathbf{z}_q(\mathcal{X})$ through codex indexing. Finally, a pre-trained BART model translates this discrete codex representation into texts.

representations. Earlier works make use of traditional methods to obtain hand-crafted representations from a sequence of EEG signals. A continuous feature representation is extracted using methods such as statistical features [53], correlation coefficient matrix from fast Fourier transform (FFT) components [43, 51, 50], wavelet transform features [18, 32, 54], and Mel-frequency cepstral coefficients (MFCCs) [5]. These hand-crafted representations can be used to establish a mapping between EEG segments and words using distance-based methods such as linear discriminative analysis (LDA) or Support Vector Machine (SVM).

On the other hand, deep learning methods, such as [52], utilize a convolutional neural network (CNN) and a Long Short-Term Memory (LSTM) to learn deep features and perform classification of user's instructions. However, the direct representation is often insufficient to discriminate a larger number of imagined text categories. In [39], EEG signals are first classified into six phonological categories as the intermediate state using a deep convolutional autoencoder. Then latent features are used as input for another encoder to predict a total of 11 speech tokens. Recently, [48] proposed a framework that uses a multi-layer transformer encoder to project word-level EEG feature sequences into EEG embeddings. Then a pre-trained BART model is used to decode these embeddings into words.

## 3 Method

The overall process of DeWave is illustrated in Figure 2, where the word-level or raw EEG features are vectorized into sequences embedding and are fed into the discrete codex. The language model generates translation output based on discrete codex representation.

### 3.1 Task Definition

Given a sequence of word-level EEG features $\mathcal{E}$, the aim is to decode the corresponding open-vocabulary text tokens $\mathcal{W}$. These EEG-Text pairs $\langle \mathcal{E}, \mathcal{W} \rangle$ are collected during natural reading, as defined in Section 4.1. We delve into two task settings: (1) Word-level EEG-to-Text Translation, where EEG feature sequences $\mathcal{E}$ are fragmented and re-ranked based on eye-fixation $\mathcal{F}$ aligned with each word token $\mathbf{w}$ in sequence $\mathcal{W}$; and (2) Raw EEG Waves to Text Translation, where EEG feature sequences $\mathcal{E}$ are directly vectorized into embedding sequences for translation without any event markers, a more challenging but practical real-time setting. DeWave is the pioneering work in this latter task.

### 3.2 Discrete Codex

Discrete representation is first proposed in VQ-VAE [45]. DeWave is the first work to introduce discrete encoding into EEG signal representation. The discrete representation could benefit both the word-level EEG features and the raw EEG wave translation. Introducing discrete encoding into brain waves could bring two aspects of advantages. 1) It is widely accepted that EEG features have a strong data distribution variance across different human subjects. Meanwhile, the datasets can only have samples from a few human subjects due to the expense of data collection. This severely weakened the generalized ability of EEG-based deep learning models By introducing discrete encoding, we

could alleviate the input variance to a large degree as the encoding is based on checking the nearest neighbor in the codex book. 2) The codex contains fewer time-wise properties which could alleviate the order mismatch between event markers (eye fixations) and language outputs. Meanwhile, because of this property of codex represents, DeWave is the first work that could realize the direct translation from raw EEG waves without any event marker to give the order.

**Inference:** Given the EEG waves $\mathcal{E}$, it is firsted vectorized into embedding as introduced in Section 3.3 with ($\mathcal{X} = \Theta(\mathcal{E}, \mathcal{F})$) or without ($\mathcal{X} = \Theta(\mathcal{E})$) eye fixations $\mathcal{F}$, where $\mathcal{X}$ is the embedding sequence. A codex book $\{\mathbf{c}_i\} \in \mathbb{R}^{k \times m}$ is initialized with number $k$ of latent embedding with size $m$. The vectorized feature $\mathcal{X}$ is encoded into $\mathbf{z}_c(\mathcal{X})$ through a transformer encoder. The discrete representation is acquired by calculating the nearest embedding in the codex of input embedding $\mathbf{x} \in \mathcal{X}$ as shown in Equation 1.

$$\mathbf{z}_q(\mathcal{X}) = \{\mathbf{z}_q(\mathbf{x})\}_i, \quad \mathbf{z}_q(\mathbf{x}) = c_k, \quad k = \mathrm{argmin}_j \|\mathbf{z}_c(\mathbf{x}) - \mathbf{c}_j\|_2 \tag{1}$$

Different from the original VQ-VAE which decodes the original input, Dewave directly decodes the translation output given the representation $\mathbf{z}_q(\mathcal{X})$. Given a pre-trained language model, the decoder predicts text output with $P(\mathcal{W}|\mathbf{z}_q(\mathcal{X}))$.

**Learn:** The codex is like a bridge connecting the vectorized EEG feature and the language model. Compared to learning direct EEG-to-text relation, DeWave learns a better discrete codex for the language model. It is easier to learn since We learn the discrete codex by the combination of the loss functions in three parts,

$$L = -\log(\mathrm{p}(\mathcal{W}|\mathbf{z}_q(\mathcal{X})) + \|\mathbf{sg}[\mathbf{z}_c(\mathbf{x})] - \mathbf{z}_q(\mathbf{x})\|_2^2 + \beta\|\mathbf{z}_c(\mathbf{x}) - \mathbf{sg}[\mathbf{z}_q(\mathbf{x})]\|_2^2 \tag{2}$$

where the loss maximize the log-likelihood of language outputs $\log(\mathrm{P}(\mathcal{W}|\mathbf{z}_q(\mathcal{X}))$ and minimize the distance between latent variable $\mathbf{z}$ and the codex value $\mathbf{z}$. Here, the $\mathbf{sg}$ denotes the stop gradients. The learning is robust for $\beta$ from 0.1-2.0, where we set it as 0.2 throughout the training process.

### 3.3 EEG Vectorization

**Word-Level EEG Features With Event Markers:** The EEG waves are first sliced into fragments according to the eye-tracking fixation of word sequences given in the annotation. Similar to [48], we calculate the statistical result of four frequency band filters, Theta band (5-7Hz), the Alpha band (8-13Hz), the Beta band (12-30Hz), and Gamma band (30Hz-) [27] to get the statistic frequency features of each fragment. It is noted that although different fragments may have different EEG window sizes, the statistical results are the same (embedding size 840). A multi-head transformer layer is applied to project the embedding into feature sequences with latent size 512.

**Raw EEG Waves: Self-Guided Waves to Discrete Codex** Our self-supervised EEG wave encoder transforms raw EEG signals into a sequence of embeddings [2, 56] as illustrated in Figure 3. It has two guiding principles: Self-Reconstruction, where the encoder is trained to transform and subsequently reconstruct the original waveforms from discrete codices; and Text Alignment, where the codices' encoding is semantically aligned with word vectors, fostering the development of text-aligned EEG signal representations.

For structure-wise, a conformer-based multi-layer encoder with specially designed hyperparameters is employed. The one-dimensional convolution layer processes the EEG waves to generate the embedding sequence [4], fusing the EEG channels into a unique embedding for each period. We apply bi-directional transformer attention layers to the sequence to capture temporal relations.

For the reconstruction process, a decoder transformer and transpose convolution structure then convert these discrete embeddings back into raw waves. Given the reconstruction process as $\tilde{\mathcal{X}} = \phi(\mathbf{z}_q(\mathcal{X}))$, the self-supervised loss could be modified into:

$$L_{\mathrm{wave}} = \frac{1}{n}\sum(\phi(\mathbf{z}_q(\mathcal{X})) - \mathcal{X})_n^2 + \|\mathbf{sg}[\mathbf{z}_c(\mathbf{x})] - \mathbf{z}_q(\mathbf{x})\|_2^2 + \beta\|\mathbf{z}_c(\mathbf{x}) - \mathbf{sg}[\mathbf{z}_q(\mathbf{x})]\|_2^2, \tag{3}$$

where the model calculates the mean square loss between the reconstructed wave and the wave ground truth to perform self-supervised training.

To obtain a semantically coherent codex, we introduce a cross-modality contrastive learning approach distinct from prevalent methods. Unlike CLIP [35, 23, 22] that contrasts CLS embeddings between

---

[4]perception field is roughly 200ms with overlap 100ms for each embedding

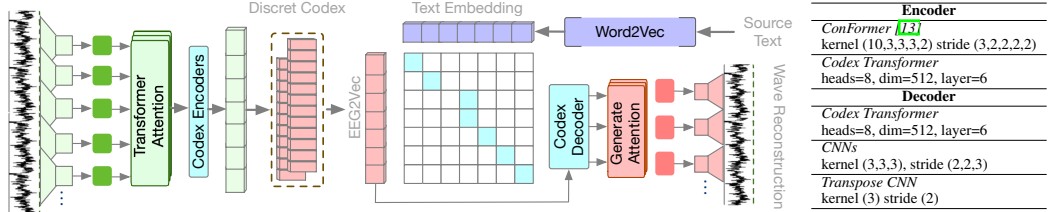

Figure 3: The image demonstrates the process of self-supervised pre-training for raw waves. The left subgraph details our strategy for directing the encoder, utilizing both self-reconstruction and text alignment through contrastive learning.

sample pairs within a mini-batch, our approach operates within a single EEG-text pair. We contrast the EEG-tokenized codex embeddings sequence $z_q$ with the text embeddings sequence $z_t$. Assuming the raw wave feature extractors can produce a token sequence in an **organized chronological order**, we treat the diagonal EEG codex and text word2vec encoding pairs as positive pairs within the sequences. All other pairs are considered negative. The model is trained to minimize the distance between embeddings of positive pairs and maximize that of the negative pairs. For a given EEG-text pair $(i, j)$, the loss is defined in Equation 4

$$L_{\text{contrast}} = -\frac{1}{n} \sum log \left[ \frac{\exp(\mathbf{s}_{ii}/\tau)}{\sum_{k=1}^{N} \exp(\mathbf{s}_{ik}/\tau)} \right], \mathbf{s}_{ij} = \mathbf{z}_q(\mathbf{x}^i)^T \mathbf{z}_t(j) \tag{4}$$

Here, $\tau$ is the temperature parameter, $N$ is the total number of EEG-text pairs in a batch, and the sum in the denominator is over all $N$ EEG-text pairs and $N$ text-EEG pairs. The EEG embeddings are expected to correctly match with their corresponding text while distinguishing them from mismatched EEG-text pairs. The total loss then becomes a combination of the original Wave2Vec loss and the contrastive loss as $L_{\text{total}} = L_{\text{wave}} + \alpha L_{\text{contrast}}$.

By this means, the model not only learns to reconstruct the EEG signal but also learns a robust representation of the signal that aligns with the corresponding text embeddings. This cross-modal learning can potentially improve the translation system by bridging the gap between EEG signals and the semantic content of the text.

## 3.4 Language Model

We used large-scale text corpus pre-trained BART [21] as the generative language model for translation output. As the EEG-to-text translation data is quite limited, leveraging the BART model could introduce prior knowledge of text relations. In that case, the translation system only needs to learn a codex representation for the language model, which is easier to learn. The codex representations are fed into pre-trained [5] BART model and get the output hidden states. A fully connected layer is applied on the hidden states to generate English tokens from pre-trained BART vocabulary $\mathcal{V}$.

## 3.5 Training Paradigm

DeWave is trained through a multi-stage process, where the training process is illustrated in Appendix C. In the first stage, we do not involve the language model in weight updates. The target of the first stage is to train a proper encoder projection $\theta_{codx}$ and a discrete codex representation $\mathcal{C}$ for the language model. In the second stage, the gradient of all weights, including language model $\theta_{BART}$ is opened to fine-tune the whole system.

## 4 Experiments

### 4.1 Dataset

DeWave utilize both ZuCo 1.0 [15] and 2.0 [17] for experiments. The dataset simultaneously recorded the text and EEG corpus during Normal Reading (NR) and Task-Specific Reading (TSR) tasks. The

---

[5] https://huggingface.co/facebook/bart-large

EEG waves are collected with a 128-channel system under a sampling rate 500Hz through a frequency band filter from 0.1Hz to 100Hz. However, after the noise canceling process, only 105 channels [15] are used for translation. Similar to [48], we slice the EEG wave according to the eye fixation and calculate the frequency features. For raw EEG waves, the signal is normalized into a value range of 0-1 for decoding. The reading task's data are divided into the train (80%), development (10%), and test (10%) respectively by 10874, 1387, and 1387 unique sentences with no intersections. Please refer to Appendix A for a detailed description.

## 4.2 Implementation Details

For word-level EEG features, we use the 56 tokens each with an 840 embedding size. For raw EEG waves, we clip or pad the EEG waves up to sample point 5500 with a constant value of zero. A transformer layer with head number 8 and a $1 \times 1$ convolutional layer are combined to fuse multiple EEG channels into an embedding sequence with size 512. DeWave uses a codex with size 2048 where each codex latent is an embedding with size 512. The ablation study (Section 4.6) gives a discussion about the codex size. All models are trained on Nvidia V100 and A100 GPUs. For the self-supervised decoding for raw waves, we use a learning rate of 5e-4 and a VQ coefficient of 0.25 for training 35 epochs. For training the codex (stage 1), DeWave uses a learning rate of 5e-4 for 35 epochs. For finetuning the translation (stage 2), DeWave uses a learning rate of 5e-6 for 30 epochs. We use the SGD as the optimizer for training all the models. Due to limited space, refer to Appendix B for more details.

## 4.3 Evaluation Metrics

We evaluate translation performance using NLP metrics, BLEU and ROUGE, as shown in Table 1. For word-level EEG features, we compare our results to EEG-to-Text [48], maintaining a consistent language model for fairness. In the absence of methods for raw EEG waves, we establish a baseline EEG-to-Text[†] by segmenting the entire EEG waves into a sequence embedding using a 200ms time window with a 100ms overlap. We adapt Wave2Vec, originally developed for speech recognition, to brain waves and compare it with our approach, DeWave. Furthermore, we adapt unsupervised raw EEG waves classification methods BENDR [11] and SCL [10], using SSL pre-training and feature extraction for comparison, underscoring the impact of discrete encoding.

Table 1: Evaluation metrics of EEG-to-Text translation under both word-level features input and raw waves input, where +Contrastive denotes boost encoder with $L_{contrast}$ mentioned in Sec. 3.3. For a fair comparison, these results keep the same teacher-forcing evaluation setting as EEG-to-Text [48].

| Source | Method | BLEU-N (%) | | | | ROUGE-1 (%) | | |
|---|---|---|---|---|---|---|---|---|
| | | N=1 | N=2 | N=3 | N=4 | R | P | F |
| **Word-level features** | EEG-to-Text [48] | 40.12 | 23.18 | 12.61 | 6.80 | **28.84** | 31.69 | 30.10 |
| | DeWave | **41.35** | **24.15** | **13.92** | **8.22** | 28.82 | **33.71** | **30.69** |
| **Raw waves** | EEG-to-Text[†] [48] | 13.07 | 5.78 | 2.55 | 1.10 | 15.22 | 18.08 | 16.36 |
| | Wave2Vec [2] | 18.15 | 8.94 | 3.89 | 2.04 | 18.96 | 23.86 | 20.07 |
| | BENDR [11] | 18.48 | 9.16 | 4.05 | 2.15 | 19.03 | 25.22 | 21.18 |
| | DeWave | **20.51** | **10.18** | **5.16** | **2.52** | **21.18** | **29.42** | **24.27** |
| | DeWave+Contrastive | **21.09** | **10.69** | **5.88** | **3.04** | **22.01** | **29.95** | **24.68** |

**Word-Level EEG Features:** For the word-level EEG feature, we observe that introducing the discrete brain representation could help DeWave reach BLEU-$\{1, 2, 3, 4\}$ scores of $41.35, 24.15, 13.92,$ and $8.22$, which respectively outperform the previous baseline by $1.23$ $(+3.06\%)$, $0.97$ $(+4.18\%)$, $1.31$ $(+10.38\%)$ and $1.54$ $(+18.73\%)$. It is observed that the increasing ratio is more significant for larger grams evaluation. DeWave achieve ROUGE-1 score $28.84$ (R), $31.69$ (P), and $30.10$ (F) which outperforms the previous baseline by $-0.02$ $(-0.06\%)$, $1.98$ $(+6.35\%)$, and $0.59$ $(+1.9\%)$. The increase of the matrix is higher for longer phrases (3-gram and 4-gram). This is rational since the single-word level EEG features are sliced by eye fixation during reading, which may naturally contain noises since the human subject may not really think about the corresponding words when looking at them.

**Raw Waves:** DeWave represents an unprecedented endeavor in translating raw EEG waves directly, obviating the need for event markers. This is achieved through the application of a learned Wave2Vec model, which converts waves into discrete codex tokens. When benchmarked against

the baseline method - which merely slices the wave according to a time window - DeWave exhibits marked improvement. It attains BLEU-1, 2, 3, 4 scores of 20.51, 10.18, 5.16, and 2.56, respectively, surpassing the baseline by margins of 7.44 (+56.92%), 4.44 (+76.1%), 2.61 (+102.35%), and 1.42 (+129.09%). This underscores the superiority of a learned discrete embedding representation over the rudimentary time-window baseline. In comparison with cutting-edge self-supervised learning (SSL) methods, DeWave consistently outperforms them, including contrastive learning (CL)-based methods such as SCL [10], the original Wave2Vec [2], and BENDR [11].

## 4.4 Cross-Subject Performance

Cross-subject performance is of vital importance for practical usage. To further illustrate the performance variance on different subjects, we train the model by only using the data from subject YAG and test the metrics on all other subjects. The results are illustrated in Figure 4, where the radar chart denotes the performance is stable across different subjects. Please refer to Appendix G for more detailed results.



Figure 4: The cross-subjects performance on ZuCo dataset.

## 4.5 Generated Samples

In Table 2, we display visualized examples of text generated from unseen EEG signals. Despite the challenge of thought translation and limited prior research, our model yields meaningful results, aligning keywords and forming similar sentence structures, although it may not yet match traditional language translation tasks.

The model is more adept at matching verbs than nouns. For instance, **implies**" vs. **implies**", **the author who wrote it**" vs. **the man who wrote it**" both effectively convey the intended sentiment of the sentence. However, when it comes to nouns, we observe a tendency towards synonymous pairs rather than precise translations, such as the man" vs. the author", Burroughs" vs. Heroughs", edition" vs. version". Our analysis suggests two potential causes for this. First, when the brain processes these words, semantically similar words might produce similar brain wave patterns. Given the inherent noise in brain waves, the codex might group these features under the same value. Second, the volume of EEG-to-Text pairs available for training is significantly smaller than that for traditional language translation. Hence, some degree of error in translating unseen nouns or sentences is to be expected.

Translation on raw EEG waves is naturally harder than word-level translation, as it lacks eye fixation to suggest the relationship between the period of waves and the word target. Table 2 meet our expectation that the results on raw EEG waves are not as good as those on word-level features, especially on the real semantic meaning of the sentence (sample (2) and sample (5) have the same target for comparison). However, the translation still could output the correct translation of certain words, such as "**much of**" vs. "**much of**", "**individual**" vs. "**individual**", and "more complicated story" vs. "more exciting thing". Although EEG-to-text is a hard topic, DeWave suggests the feasibility of translation improvements.

## 4.6 Ablation Study

**Discrete Codex** DeWave encodes EEG waves into a discrete codex, aiming for a language model-friendly representation. We evaluated performance against varying codex sizes (1024 to 8192) to ascertain if larger sizes yield better results. As depicted in Figure 5, we found no strong correlation between codex size and model performance. A codex size of 2048 yielded the highest BLEU score average, and while the ROUGE score slightly improved with larger sizes, there was no clear evidence that increasing codex size consistently enhanced performance.

Table 2: Translation results on the unseen EEG waves, where **bold** denotes a correct match between ground truth and our prediction. Underline denotes a fuzzy match with similar semantic meanings. For a fair comparison, these results keep the same teacher-forcing evaluation setting as EEG-to-Text [48]. This means the decoding process eliminates accumulated errors and just predicts the current token with the GT token from the last step.

| | **Decoding Results with Eye Fixation Assistance** |
|---|---|
| (1) | Ground Truth: Everything its title **implies**, a standard-**issue** crime drama spat out from the Tinseltown assembly line... |
| | Prediction: is own **implies**, including great of **issue**, novel of the beginningseltowns.... |
| (2) | Ground Truth: "The Kid Stays in the Picture" is a great story, terrifically told by **the man** **who wrote it** but this Cliff Notes edition **is a cheat**. |
| | Prediction: The film "says in the Game" is a film about but movie was written, **the author** **who wrote it**. also its version **is a cheat**. |
| (3) | Ground Truth: During **Kerouac's time** at Columbia University, **Burroughs and Kerouac** got into trouble with the law for failing to report **a murder**; this incident **formed** the **basis of** a mystery novel ... |
| | Prediction: **Keouac's time** at the, , **Heroughs and Kerouac** were along a for the police for their to pay **the murder**. they **led led** **the basis of** the lawsuit ... |
| | **Decoding Results with Raw Waves** |
| (4) | Ground Truth: Every **individual** will see the movie through the prism of his or her own beliefs and prejudices ... |
| | Prediction: Everyday **individual** is their results. their eyes of his or her own personal. desires. and ... |
| (5) | Ground Truth: "The Kid Stays in the Picture" is a great story, terrifically told by the man who wrote it but this Cliff Notes edition is a cheat. |
| | Model Output: The Price'ssays in the Middle. and a common deal. and for good. this moment. made it is still little. |
| (6) | Ground Truth: **much of** this well-acted but dangerously slow thriller feels **like** a preamble to a bigger, more complicated story, |
| | Model Output: **much of** this is-being but not over-. **like** it disaster-ble to a new and more exciting thing, |

Raw EEG wave performance fluctuates noticeably with codex size. Performance improves when increasing the codex size from 1024 to 2048, but any larger size reduces performance. This variation may result from the training formation; word-level EEG data, selected by eye fixations, contains less noise than raw waves. We hypothesize that our current training data may be insufficient for larger codex sizes. Additionally, our experiments indicate that the latent dimension of the codex doesn't significantly affect performance.

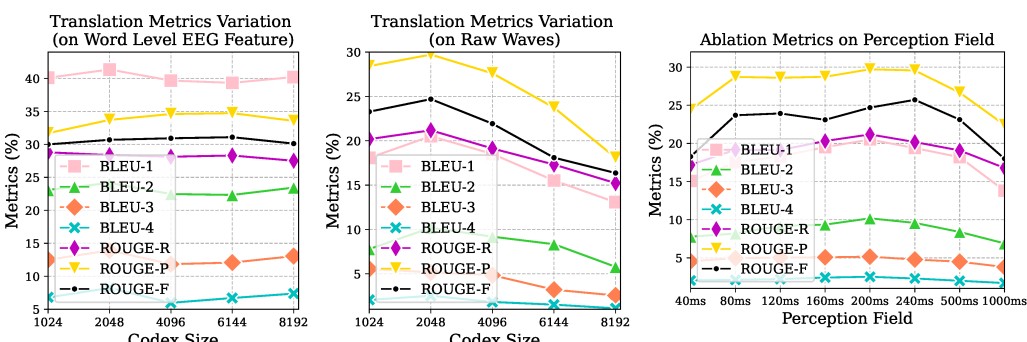

Figure 5: Ablation study on different codex sizes and perception fields (raw waves).

**Perception Time Window** We also conduct the ablation study on the model structure for the wave2vec model illustrated in Figure 5. As the model utilize a multi-layer CNN model to slide through the raw waves, the model compresses the waves for perception. The compress ratio decided how large the perception field is for each extracted embedding feature. As described in Section 3.3, the model utilized a perception field of 200ms with an overlap of 100ms. We conduct an ablation study of different perception fields and report it in Figure 5 Where it is observed that the model performance is significantly lower when the perception field is smaller than 80ms or larger than 240ms. The model could achieve similar results in the perception field 120ms to 240ms. The model reaches a small peak around 200ms to 240ms. We think this phenomenon is rational since the normal reading speed for humans is around 160-400 words per minute (reading speed may vary from different

material and human subjects). In other words, the reading period for each word is 150-375ms on average, which roughly meets our observation between 200ms-240ms.

**Self-Supervision Initialization**   Theoretically, we could pre-train the codex by introducing a decoder and calculate the reconstruction loss with the original input for both word-level EEG and raw EEG waves. We prefix other parameters as our best setting with codex size 2048 and compare the impact in Table 3.

Table 3: Ablation on self-supervised pre-trained codex weights.

| | Pretrain | BLEU (%) | | | ROUGE-1 (%) | | |
|---|---|---|---|---|---|---|---|
| | | N=1 | N=2 | N=3 | R | P | F |
| **Word-level** | ✓ | 41.35 | 24.15 | 13.92 | 28.82 | 33.71 | 30.69 |
| **features** | × | 40.71 | 22.94 | 12.40 | 28.01 | 34.08 | 30.63 |
| **Raw waves** | ✓ | 20.51 | 10.18 | 5.16 | 21.18 | 29.42 | 24.27 |
| | × | 16.58 | 7.78 | 3.68 | 17.86 | 19.84 | 18.33 |

For word-level EEG features, the impact of pre-train is mostly on BLEU scores while the ROUGE score does not have much variance. Without pre-train, the BLEU-$\{1, 2, 3\}$ respectively drop by 0.64, 1.21, and 1.52. For direct translation on raw waves, the impact is significantly larger. Without self-supervised initialization, the BLEU-$\{1, 2, 3\}$ respectively drop by 3.93 ($\downarrow 19.16\%$), 2.40 ($\downarrow 23.57\%$), and 1.48 ($\downarrow 28.68\%$). Similar observation also appears on ROUGE scores. This phenomenon is rational since raw wave decoding requires the model to pick useful features without any help from eye fixations. The self-supervised initialization could help the model form a preliminary ability to extract time-wise or channel-wise features from raw waves.

## 5   Limitations

Despite DeWave's enhancements in EEG-to-Text translation using a discrete codex and raw wave encoding, its accuracy remains far from real-life scenarios compared to traditional language-to-language translations. Also, to keep a fair comparison with EEG-to-Text, this paper uses a teacher-forcing setting in evaluation. This setting eliminates accumulation error and turns the sequence decoding task into a word-level classification task given the ground truth token from the previous step. This setting is relatively easier yet we think it is still valuable as it could suggest feature extraction quality while keeping a fair comparison.

Additionally, the experiments in this paper are restricted to public neural reading data, not fully aligning with the "silent speech" concept of direct thought translation from human brains. Instead, the current ZuCo dataset is collected by giving people reading stimuli. This paper focuses on introducing Wav2Vec formation of feature extraction on raw EEG waves and introduces discrete codex as learnable representations for the EEG-to-Text translation domain. One scientific problem in this domain is a better way of doing the "silent speech" task, which we are exploring as on-going research.

## 6   Conclusion

This paper presents DeWave, a framework for the recently proposed open-vocabulary EEG-to-Text translation task [48], introducing the concept of discrete codex encoding. This approach brings enhancement in corpus text relevancy metrics, such as BLEU and ROUGE. DeWave also expands the task to decode raw EEG waves without the assistance of eye fixation markers. Despite these advancements, the quality of brain decoding results remains substantially low and remains teacher forcing setting for fair comparison. The translation of thoughts directly from the brain is a valuable yet challenging endeavor that warrants significant continued efforts. In our ongoing work, we are exploring more reasonable settings that remove teacher forcing for both training and testing. We will also include the "neural-feedback" mechanism in this EEG-to-Text research to enhance the scientific value of this domain.

## Acknowledgement

This work was supported in part by the Australian Research Council (ARC) under discovery grants DP210101093 and DP220100803, and the GrapheneX-UTS Human-Centric AI Centre sponsored by GrapheneX (2023-2031).

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
