# Supplementary Material for DeWave:
# Discrete Encoding of EEG Waves for EEG to Text Translation

In this material, we will give more technical details as well as additional experiments to support the main paper. The overview of the proposed framework, DeWave, is illustrated in Figure 6.

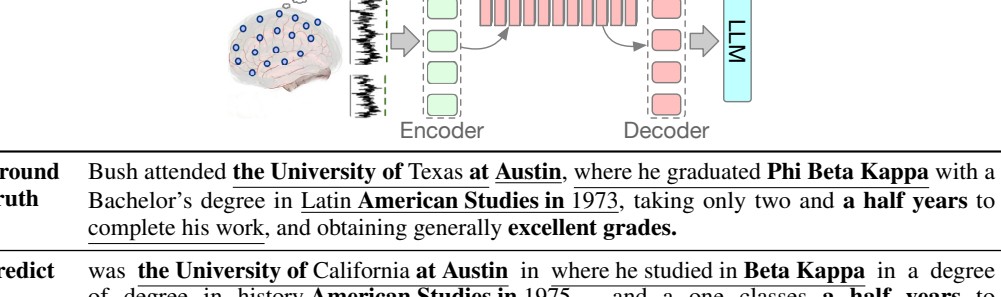

| | |
|---|---|
| **Ground Truth** | Bush attended **the University of** Texas **at** Austin, where he graduated **Phi Beta Kappa** with a Bachelor's degree in Latin **American Studies in** 1973, taking only two and **a half years** to complete his work, and obtaining generally **excellent grades.** |
| **Predict** | was **the University of** California **at Austin** in where he studied in **Beta Kappa** in a degree of degree in history **American Studies in** 1975. and a one classes **a half years** to complete the degree. and was a **excellent grades.** |

Figure 6: Overall illustration of translating EEG waves into text through quantised encoding.

## A  Dataset

ZuCo stands for Zurich Cognitive Language Processing Corpus (ZuCo), a dataset that includes both raw and preprocessed eye-tracking and electroencephalography (EEG) data. The data is collected by having human subjects read given text corpora while simultaneously recording both their eye-tracking signals and EEG waves. The recording is done using the Biosemi-128 system, which, after denoising, provides 105 out of 128 channels for downstream tasks. The dataset comprises two versions: ZuCo 1.0, collected from 12 subjects, and ZuCo 2.0, collected from 18 subjects [15, 17].

The text corpora within the ZuCo dataset are sourced from a diverse set of textual genres, including 1) Wikipedia articles, 2) movie reviews, and 3) the BNC (British National Corpus). This diversity ensures a wide variety of syntactic structures and word frequencies. The dataset records data during two tasks: Normal Reading (NR) and Task-Specific Reading (TSR). In our experiments, DeWave utilizes both ZuCo 1.0 [15] and 2.0 [17]. The EEG features are captured using a 128-channel system with a sampling rate of 500Hz, filtered through a frequency band ranging from 0.1Hz to 100Hz. After noise canceling, only 105 channels are deemed suitable for translation [15].

For word-level EEG feature translation, eye-fixation data associated with each word during reading is available in the ZuCo dataset. Following the approach similar to [48], we extract segments of the EEG wave according to eye fixations. Words fixated upon multiple times have their EEG fragments concatenated for processing. To process these word-level EEG features, we compute statistical results across four frequency band filters: the Theta band (5-7Hz), the Alpha band (8-13Hz), the Beta band (12-30Hz), and the Gamma band (30Hz and above) [27]. Consequently, the feature size for each word totals $105 \times 4 \times 2 = 840$. For raw EEG waves, the signals are normalized to a range between 0 and 1 for decoding.

The dataset is split into training (80%), development (10%), and testing (10%) sets, comprising 10,874, 1,387, and 1,387 unique sentences, respectively, with no overlap. We further conducted a statistical analysis on the sentences extracted from the dataset, details of which are reported below.

Table 4: Statistical analysis of sentences from the ZuCo dataset.

| Feature | ZuCo 1.0 Natural Reading | ZuCo 2.0 Natural Reading |
|---|---|---|
| Sentences | 300 | 390 |
| Sent. length | $21.3 \pm 10.6$ | $19.6 \pm 8.8$ |
| Total words | 6386 | 6828 |
| Word length | 6.7 | 4.9 |

## B  Implementation Details

We release our implementation code through GitHub to contribute to this area. Currently the basic code are available through an anonymous link[6] For word-level EEG features, we use the $56$ tokens each with an $840$ embedding size. The codex encoder for word-level features is a 6-layer transformer encoder with head number 8, hidden embedding 512. For raw EEG waves, we clip or pad the EEG waves up to sample point 5500 with a constant value of zero, which scales up to 11 seconds according to the sampling rate of 500 Hz. The codex encoder for raw EEG wave features is illustrated in Section 3.3, where a 6-layer CNN encoder slides through the whole wave and gets the embedding sequence. A transformer layer with head number 8 and a $1 \times 1$ convolutional layer are combined to fuse multiple EEG channels into one embedding with size 512.

The codex encoder shares the same structure with word-level features. DeWave uses a codex with size 2048 where each codex latent is an embedding with size 512. The ablation study gives a discussion about the codex size. All models are trained on Nvidia V100 and A100 GPUs. For the self-supervised decoding for raw waves, we use a learning rate of 5e-4 and a VQ coefficient of 0.25 for training 35 epochs. For training the codex (stage 1), DeWave uses a learning rate of 5e-4 for 35 epochs. For finetuning the translation (stage 2), DeWave uses a learning rate of 5e-6 for 30 epochs. We use the SGD as the optimizer for training all the models.

## C  Training Paradigm

DeWave is trained through a multi-stage process, where the training process is illustrated in Appendix algorithm 1. Before the two-stage training, if the input of the model is the raw waves, we initialize the wave2vec model with a self-supervised pre-training described in section 3.3. The self-supervised training is realized by encoding raw waves into discrete codex and reconstructing the discrete codex into original raw waves. The training process for self-supervised initialization utilizes the SGD algorithm with a learning rate of 0.0005 for 30 epochs with 0.1 times the learning rate decrease at epoch 20. In the first stage, we do not involve the language model in weight updates. The target of the first stage is to train a proper encoder projection $\theta_{codx}$ and a discrete codex representation $\mathcal{C}$ for the language model. Intuitively, if the learning of the codex is successful, the translator could receive a better representation that is closer to the original representation, word2vec embedding. The training for the first stage is optimized by the SGD optimizer with a learning rate of 0.0005 for 35 epochs. However, the language model is trained on word tokens, which may not be perfectly suitable for brain tokens. In the second stage, the gradient of all weights, including language model $\theta_{BART}$ is opened to fine-tune the whole system. The training for the second stage is optimized by the SGD optimizer with a learning rate of $1e - 6$ for 35 epochs.

## D  Codex Visualization

Since the logic is to learn a discrete codex from brain dynamics, it naturally arises whether the codex value distribution between raw waves and frequency features has differences. In that case, we conduct additional experiments to visualize the learned codex book with T-SNE methods and report the results in Fig 7. Ideally, the purpose of the discrete codex is to make the language model have a better understanding of brain encoding. In that case, the learned codex regardless of whether it is for frequency features or raw waves, should be approximately the same as the word2vec embedding. In other words, the distribution of the codex should be similar. Fig. 7 supports our expectation,

---

[6] https://github.com/duanyiqun/DeWave

**Algorithm 1** Training procedure

**Input**: EEG $\mathcal{E}$, Vocabulary $\mathcal{V}$, Marker$\mathcal{F}$, Target $\mathcal{W}$
**Parameter**: Codex $\mathcal{C} = \{\mathbf{c}\}, \theta_{codex}, \theta_{BART}, \Theta_{wave}$

1: **if** decode raw waves **then**
2:     $\underset{\mathcal{C},\theta_{codex},\Theta_{wave}}{\arg\min}$ $L_{\text{wave}}$
3:     Vectorize $\mathcal{X} = \Theta(\mathcal{E})$
4: **else**
5:     Vectorize $\mathcal{X} = \Theta(\mathcal{E}, \mathcal{F})$
6: **end if**
7: Stage-1: Train codex
8: **while** Iteration steps **do**
9:     $\underset{\mathcal{C},\theta_{codex}}{\arg\min} L(\mathcal{X})$
10: **end while**
11: Stage-2: Finetune Language Model
12: **while** Iteration steps **do**
13:     $\underset{\mathcal{C},\theta_{codx},\theta_{BART}}{\arg\min}$ $L(\mathcal{X})$
14: **end while**
15: **return** $\mathcal{C}, \theta_{codex}, \theta_{BART}, \Theta_{wave}$

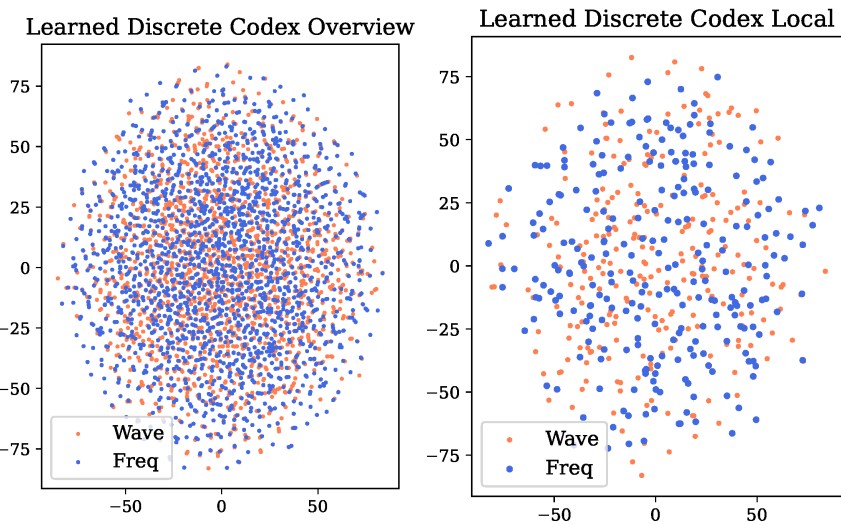

Figure 7: Visualization of codex value distribution, where the left is the global distribution, and the right is the local distribution.

where the codex learned from frequency and raw waves have very similar distributions. However, the frequency codex has more coverage with corner cases and boundaries. We think this is rational since frequency features are naturally easier to distinguish as it has introduced manually selected features. The performance gap between raw waves and frequency features supports this point as well. Still, the similarity of the distribution illustrates the rationality of the learned codex.

# E   Motivation and Preliminary Tests with LLMs

We provide additional insights into our experiments with larger language models. While our primary experiments utilized BART to ensure consistency in the decoder scale with prior works, it was paramount for us to ascertain that the observed improvements emanated from discrete coding and not just a more sophisticated decoder.

The potential of bridging brain activities with larger language models (LLMs) and advancing towards AGI is a significant research avenue. Recognizing this, we recently undertook an ablation study, where we replaced the BART decoder with OPT and Llama V1. Contrary to our expectations, the performance enhancement was modest. Given the vast implications of this area, we previously refrained from including these findings in the main manuscript for reasons of prudence.

Limited by our computational resources, we employed the PyTorch FSDP mode to fine-tune the OPT-1.3B and Llama-1 7B models with half-precision across three epochs. Taking cues from Mini-GPT4's method for handling visual tokens, the tokenized EEG waves were prompted into LLMs. The performance metrics for our experiments with BART, OPT 1.3B, and Llama-1 7B decoders are tabulated below:

Table 5: Performance metrics for different decoders on the ZuCo dataset.

| Source | Decoder | BLEU-1 | BLEU-3 | ROUGE-R | ROUGE-P | ROUGE-F |
|--------|---------|--------|--------|---------|---------|---------|
| Word-level features | DeWave | 41.35 | 13.92 | 28.82 | 33.71 | 30.69 |
| Word-level features | DeWave + OPT 1.3B | 41.97 | 14.06 | 28.98 | 33.82 | 30.86 |
| Word-level features | DeWave + Llama-1 7B | 42.84 | 15.03 | 29.42 | 35.43 | 32.05 |
| Raw Waves | DeWave | 20.51 | 5.16 | 21.18 | 29.42 | 24.27 |
| Raw Waves | DeWave + OPT 1.3B | 21.31 | 5.84 | 22.09 | 29.94 | 25.42 |
| Raw Waves | DeWave + Llama-1 7B | 22.05 | 6.03 | 22.45 | 30.01 | 26.08 |

From the table, it's evident that while the LLMs offer some enhancement, the gains are not as pronounced as one might expect. This underscores the complexity of the problem and the challenges of bridging brain activities with LLMs. This simple experiment provides a deeper dive into our experiments with larger language models. We believe these findings offer additional perspectives and pave the way for more nuanced research in this domain.

## F   Generated Samples

In this section, we visualize the generated decoding text results on brain waves and compare them with the ground truth in Table 6 and Table 7. It suggests that the results are even better on long and simple sentences. For example, even for the ground truth with long and logic as below, the prediction could still match key information throughout the whole sentence.

---

Ground Truth:
Bush attended **the University of** Texas **at  Austin**,  where he graduated **Phi Beta Kappa** with a Bachelor's degree in Latin **American Studies in** 1973, taking only two and **a half years** to complete his work, and obtaining generally **excellent grades.**

---

Model Output:
was **the University of** California **at Austin**  in  where he studied in **Beta Kappa** in a degree of degree in history **American Studies in** 1975. and a one classes **a half years** to complete the degree. and was a **excellent grades.**

---

People is feasible to guess the meaning of a human based on the translation from brain waves. In the example above, the model recognizes through waves that **the University of** xxx at Austin. Although it is a factual mistake that the University of California is not at Austin, it still suggests that the model could approximately capture the semantic meaning through non-invasive brain waves. A similar observation applies that the model recognizes that it is the xx **American Studies in** xx years however the model predicts history **American Studies in** 1975 rather than the ground truth is Latin **American Studies in** 1973. Surprisingly, even the years have correlations at this stage.

Table 6: Translation comparison between the ground truth and the prediction on brain waves with eye fixation on task v2.0 dataset.

| | |
|---|---|
| (1) | Ground Truth: The book was awarded the 1957 Pulitzer Prize for Biography ... |
| | Prediction:first is published the Pulitzer Pulitzer Prize for Literatureography ... |
| (2) | Ground Truth: Kennedy's other decorations of the Second World War include the Purple Heart, Asiatic-Pacific Campaign Medal, and the World War II Victory Medal. |
| | Prediction: eth was son son were the day World War were a famous Heart and thepenatic StarAmerican,,, and the American War II Victory Medal. |
| (3) | Ground Truth: In 1958, Kennedy published the first edition of his book A Nation of Immigrants, closely following his involvement in the Displaced Persons Act and the 1957 bill to bring families together. |
| | Prediction: the, the was his novel of of his autobiography, Life of Millionsigrants, which followed the experiences in the Vietnamrael Persons Movement of the Civil assassination of abolish it together. |
| (4) | Ground Truth: After World War II, Kennedy entered politics (partly to fill the void of his popular brother, Joseph P. Kennedy, Jr., on whom his family had pinned many of their hopes but who was killed in the war) ... |
| | Prediction: the War II, the was the asasly as avoid a void left a father father, John Kennedy. Kennedy), who.) who the he father had been the hopes the hopes). who had assassinated in the war. ... |
| (5) | Ground Truth: In 1946, Representative James Michael Curley vacated his seat in an overwhelmingly Democratic district to become mayor of Boston and Kennedy ran for that seat, beating his Republican opponent by a large margin. |
| | Model Output: the, the John W Smithley was the seat in the unsuccessful Republican Congress of become a of New. become's for president office in which incumbent opponent opponent, a landslide margin. |
| (6) | Ground Truth: He was reelected twice, but had a mixed voting record, often diverging from President Harry S. Truman and the rest of the Democratic Party. |
| | Model Output: was a- to in in lost to less record record. and votingting from the Obama Truman. Truman's his Republican of the Republican Party. |
| (7) | Ground Truth: He was reelected twice, but had a mixed voting record, often diverging from President Harry S. Truman and the rest of the Democratic Party. |
| | Model Output: was a- to in in lost to less record record. and votingting from the Obama Truman. Truman's his Republican of the Republican Party. |
| (7) | Ground Truth:However, the U.S. Navy accepted him in September of that year. |
| | Model Output: it film.S. government has the as the. that year. |
| (8) | Ground Truth: In the spring of 1941, Kennedy volunteered for the U.S. Army, but was rejected, mainly because of his troublesome back. |
| | Model Output: the meantime of 2016, the was to the first.S. Army. and was discharged for and because he his age temper. |
| (9) | Ground Truth: When Bush was seventeen, he went to Leon, Mexico, as part of his school's student exchange program. |
| | Model Output: the was president, he was to aidas Nebraska, to a of a father's " exchange program. |
| (10) | Ground Truth: In November 1977 he was sent to the Venezuelan capital of Caracas, in South America, to open a new operation for the bank. |
| | Model Output: the,, was born to the United prison, Manacas to where a America, to work a restaurant bank. the government. |
| (11) | Ground Truth: In 1923 he was awarded the inaugural Bôcher Memorial Prize by the American Mathematical Society. |
| | Model Output: the, was born the Nobel PulitzerAFTAne Prize Medal for the French Academyical Society. |
| (12) | Ground Truth: The mathematician Garrett Birkhoff (1911-1996) was his son. |
| | Model Output: tfirst and Wkoff was1802-19) was a name. |

Table 7: Translation comparison between the ground truth and the prediction on brain waves with eye fixation on task v2.0 dataset.

| | |
|---|---|
| (13) | Ground Truth: Jeb Bush was born in Midland, Texas, where his father was running an oil drilling company. |
| | Model Output: uan Bush was a in 18way, Texas, in he father was an insurance refinery company. |
| (14) | Ground Truth: He was noted for his lyrical playing, and performed with John Coltrane, Dexter Gordon, Hampton Hawes, Jackie McLean, and Ike and Tina Turner, among others. |
| | Model Output: was a for his "ical, style and his a a Legendtrane in who Gordon and and Fes and and GleGovern and and others Turner Tina Turner. among others. |
| (15) | Ground Truth: He later became an educator, teaching music theory at the University of the District of Columbia; he was also director of the District of Columbia Music Center jazz workshop band. |
| | Model Output: was added a actor and and at to and the University of California Arts of Columbia. he was also a of the University's Columbia's Festival. program.. |
| (16) | Ground Truth: John Ellis "Jeb" Bush (born February 11, 1953), a Republican, is the forty-third and current Governor of Florida. |
| | Prediction: nie,Johnock" Ellis (19 18 17, 18) a former, was the author-six president final governor of Texas. |
| (17) | Ground Truth: He is a prominent member of the Bush family, the younger brother of President George W. Bush and the second son of former President George H. W. Bush and Barbara Bush. |
| | Prediction: was a former member of the American family and and first brother of President Bush Bush. Bush. the father son of President President Richard H. W. Bush. his Bush. |
| (18) | Ground Truth: After earning his degree, Bush went to work in an entry level position in the international division of Texas Commerce Bank, which was run by Ben Love. ... |
| | Prediction: the his degree, he was on work for the office- position in the Department banking of the A.. where was later by his Carsonll ... |
| (19) | Ground Truth: Following the 1980 presidential election, Bush and his family moved to Miami-Dade County, Florida. |
| | Model Output: the deaths election, the was his wife moved to California,Dade County, Florida, |
| (20) | Ground Truth: He took a job in real estate with Armando Codina, a 32-year-old Cuban immigrant and self-made American millionaire. |
| | Model Output: was a liking as the estate in aando Iino in a former-year-old from immigrant from former-made millionaire businessman. |
| (21) | Ground Truth: [4] Situated in Liberty City, Dade County, the school is located just outside of greater Miami, in an area plagued by poverty. |
| | Model Output: ..]] Theuations in the,, Missouri. County, North city is a in outside of the New. Florida the area known by crime and |
| (22) | Ground Truth:The co-founder, working alongside Bush as a partner, was T. Williard Fair, a well-known local black activist and head of the Greater Miami Urban League. |
| | Model Output: first-founder of John with his, a consultant, was aoniJard,banks who former-known film politiciansmith and activist of the Black Chicago NAACP League. |
| (23) | Ground Truth: Governor Buddy MacKay (55% to 45%) to become governor, after courting moderate voters and Hispanics. |
| | Model Output: or of RoKay ofleft) of 55%) of the governor of and theting the opposition in winning. |
| (24) | Ground Truth: At the urging of his wife, Columba, a devout Mexican Catholic, the Protestant Bush became a Roman Catholic. |
| | Model Output: the same of his wife, hea, he young Catholic Catholic, he actor pastorman a Catholic Catholic in |
| (25) | Ground Truth: Bush attended the University of Texas at Austin, where he graduated Phi Beta Kappa with a Bachelor's degree in Latin American Studies in 1973, taking only two and a half years to complete his work, and obtaining generally excellent grades. |
| | Model Output: was the University of California at Austin in where he studied in Beta Kappa in a degree of degree in history American Studies in 1975. and a one classes a half years to complete the degree. and was a excellent grades. |

# G    Subject Wise Evaluation

Section 4.3 introduces subject-wise metric evaluation on word-level EEG features by removing the subject to be tested from the training data, and then training the model from scratch for testing. The results are shown in Fig 4 in the main paper, where different subjects share the same reading article. However, in supplementary details, we conduct a more detailed subject-wise evaluation in that we respectively train the model on every single subject on task v2.0 and test every subject to report the cross-subject performance. The single subject denotes the model only trained on a single subject on the task v2.0 dataset. For each subject, however, due to limited data, we add all data from task 1.0 as an assistance base. The results for each subject are reported below.

Here we seleceted subject YAC (Table 8), YAK (Table 9), YDG (Table 10), YFS (Table 11), YSL (Table 12), and YMD (Table 13) to report the performance. We visualize the metrics by clustering the same metrics value of different subjects in one radar chart. It is observed that the model performance might not be optimized if we train and test on the same subject. For example, if we train the model with task2.0 data from subject YFS and test on all subjects, the YFS subject only reaches BLUE-$\{1-4\}$ 43.32, 26.30, 14.68, and 7.62, which is lower than YDS, YAG, YRP, .. etc. which reach 43.79, 26.46, 14.94, and 8.03. This suggests the cross-subject robustness of the proposed DeWave model. Also, since we use the same visualization scale for each radar chart, the area of the chart suggests the performance level. It is observed that if we change the training data from subject to subject, the average performance of every subject is affected by a similar trend. We think this phenomenon is caused by the different signal-to-noise ratios. Some subjects might naturally have less noise interference which makes it easier for the model to learn meaningful features during the training process.

# H    Supplementatary Conclusion

In this supplementary material, we give implementation details, training schema, and most importantly, more generated results and subject-wise evaluation of the proposed DeWave model. The generated results suggest a surprisingly good correlation between the model output on brain waves and the ground truth, even in long sentences with logic. Although there are factual and cognitive mistakes in the translation, it is still feasible to guess the meaning of a human based on the translation from brain waves. The subject-wise evaluation suggests the DeWave model is stable across different human subjects. Please refer to the tables attached below.

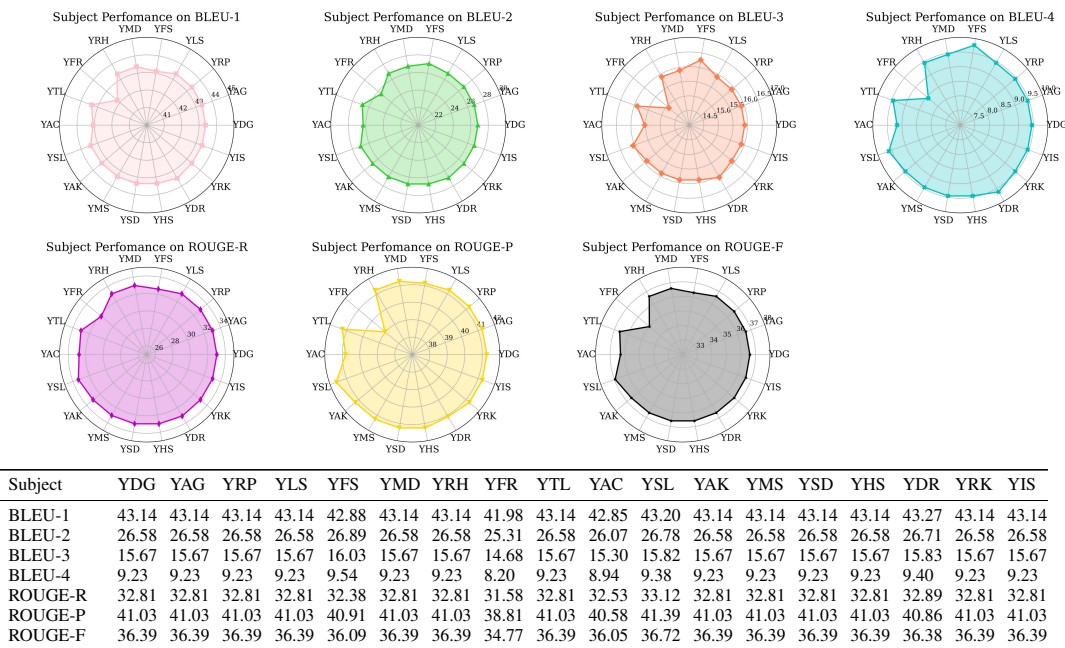

| Subject | YDG | YAG | YRP | YLS | YFS | YMD | YRH | YFR | YTL | YAC | YSL | YAK | YMS | YSD | YHS | YDR | YRK | YIS |
|---------|-----|-----|-----|-----|-----|-----|-----|-----|-----|-----|-----|-----|-----|-----|-----|-----|-----|-----|
| BLEU-1 | 43.14 | 43.14 | 43.14 | 43.14 | 42.88 | 43.14 | 43.14 | 41.98 | 43.14 | 42.85 | 43.20 | 43.14 | 43.14 | 43.14 | 43.14 | 43.27 | 43.14 | 43.14 |
| BLEU-2 | 26.58 | 26.58 | 26.58 | 26.58 | 26.89 | 26.58 | 26.58 | 25.31 | 26.58 | 26.07 | 26.78 | 26.58 | 26.58 | 26.58 | 26.58 | 26.71 | 26.58 | 26.58 |
| BLEU-3 | 15.67 | 15.67 | 15.67 | 15.67 | 16.03 | 15.67 | 15.67 | 14.68 | 15.67 | 15.30 | 15.82 | 15.67 | 15.67 | 15.67 | 15.67 | 15.83 | 15.67 | 15.67 |
| BLEU-4 | 9.23 | 9.23 | 9.23 | 9.23 | 9.54 | 9.23 | 9.23 | 8.20 | 9.23 | 8.94 | 9.38 | 9.23 | 9.23 | 9.23 | 9.23 | 9.40 | 9.23 | 9.23 |
| ROUGE-R | 32.81 | 32.81 | 32.81 | 32.81 | 32.38 | 32.81 | 32.81 | 31.58 | 32.81 | 32.53 | 33.12 | 32.81 | 32.81 | 32.81 | 32.89 | 32.81 | 32.81 |
| ROUGE-P | 41.03 | 41.03 | 41.03 | 41.03 | 40.91 | 41.03 | 41.03 | 38.81 | 41.03 | 40.58 | 41.39 | 41.03 | 41.03 | 41.03 | 41.03 | 40.86 | 41.03 | 41.03 |
| ROUGE-F | 36.39 | 36.39 | 36.39 | 36.39 | 36.09 | 36.39 | 36.39 | 34.77 | 36.39 | 36.05 | 36.72 | 36.39 | 36.39 | 36.39 | 36.39 | 36.38 | 36.39 | 36.39 |

Table 8: Subject-wise evaluation results on a model trained with subject **YAC**, where the radar chart suggests the performance variance on different subjects on each metric.

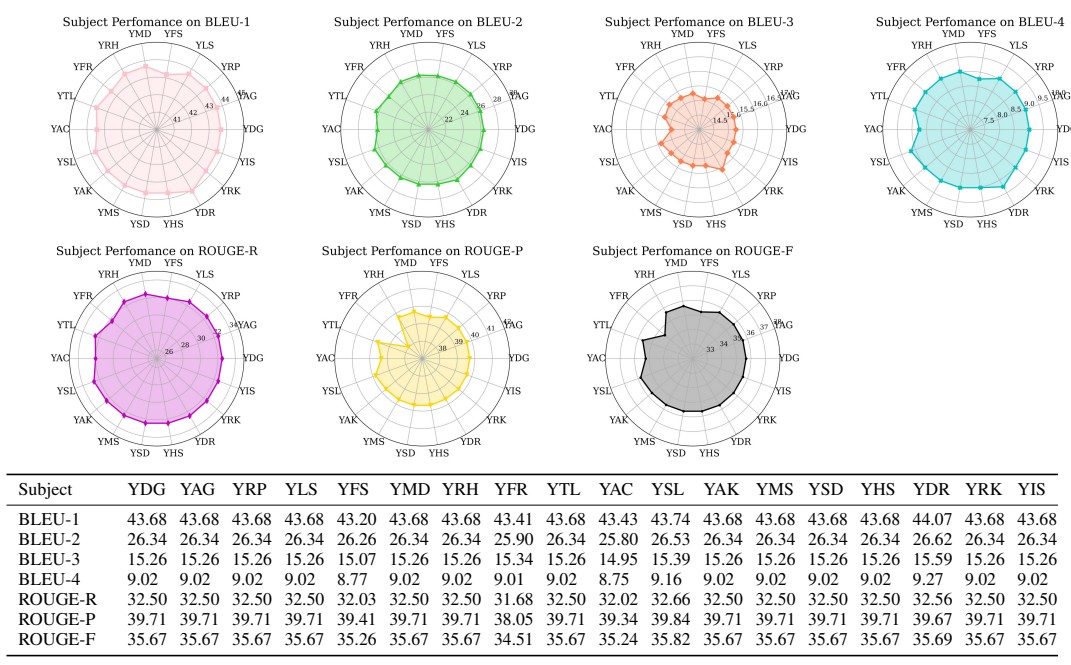

| Subject | YDG | YAG | YRP | YLS | YFS | YMD | YRH | YFR | YTL | YAC | YSL | YAK | YMS | YSD | YHS | YDR | YRK | YIS |
|---------|-----|-----|-----|-----|-----|-----|-----|-----|-----|-----|-----|-----|-----|-----|-----|-----|-----|-----|
| BLEU-1 | 43.68 | 43.68 | 43.68 | 43.68 | 43.20 | 43.68 | 43.68 | 43.41 | 43.68 | 43.43 | 43.74 | 43.68 | 43.68 | 43.68 | 43.68 | 44.07 | 43.68 | 43.68 |
| BLEU-2 | 26.34 | 26.34 | 26.34 | 26.34 | 26.26 | 26.34 | 26.34 | 25.90 | 26.34 | 25.80 | 26.53 | 26.34 | 26.34 | 26.34 | 26.34 | 26.62 | 26.34 | 26.34 |
| BLEU-3 | 15.26 | 15.26 | 15.26 | 15.26 | 15.07 | 15.26 | 15.26 | 15.34 | 15.26 | 14.95 | 15.39 | 15.26 | 15.26 | 15.26 | 15.26 | 15.59 | 15.26 | 15.26 |
| BLEU-4 | 9.02 | 9.02 | 9.02 | 9.02 | 8.77 | 9.02 | 9.02 | 9.01 | 9.02 | 8.75 | 9.16 | 9.02 | 9.02 | 9.02 | 9.02 | 9.27 | 9.02 | 9.02 |
| ROUGE-R | 32.50 | 32.50 | 32.50 | 32.50 | 32.03 | 32.50 | 32.50 | 31.68 | 32.50 | 32.02 | 32.66 | 32.50 | 32.50 | 32.50 | 32.50 | 32.56 | 32.50 | 32.50 |
| ROUGE-P | 39.71 | 39.71 | 39.71 | 39.71 | 39.41 | 39.71 | 39.71 | 38.05 | 39.71 | 39.34 | 39.84 | 39.71 | 39.71 | 39.71 | 39.71 | 39.67 | 39.71 | 39.71 |
| ROUGE-F | 35.67 | 35.67 | 35.67 | 35.67 | 35.26 | 35.67 | 35.67 | 34.51 | 35.67 | 35.24 | 35.82 | 35.67 | 35.67 | 35.67 | 35.67 | 35.69 | 35.67 | 35.67 |

Table 9: Subject-wise evaluation results on a model trained with subject **YAK**, where the radar chart suggests the performance variance on different subjects on each metric.

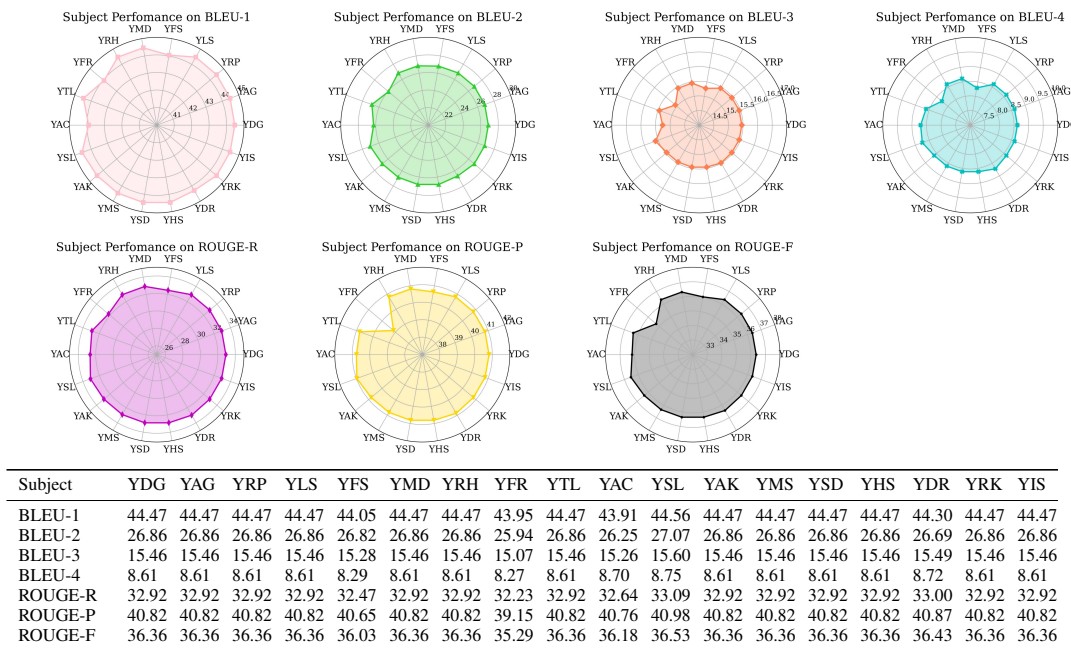

| Subject | YDG | YAG | YRP | YLS | YFS | YMD | YRH | YFR | YTL | YAC | YSL | YAK | YMS | YSD | YHS | YDR | YRK | YIS |
|---------|-----|-----|-----|-----|-----|-----|-----|-----|-----|-----|-----|-----|-----|-----|-----|-----|-----|-----|
| BLEU-1  | 44.47 | 44.47 | 44.47 | 44.47 | 44.05 | 44.47 | 44.47 | 43.95 | 44.47 | 43.91 | 44.56 | 44.47 | 44.47 | 44.47 | 44.47 | 44.30 | 44.47 | 44.47 |
| BLEU-2  | 26.86 | 26.86 | 26.86 | 26.86 | 26.82 | 26.86 | 26.86 | 25.94 | 26.86 | 26.25 | 27.07 | 26.86 | 26.86 | 26.86 | 26.86 | 26.69 | 26.86 | 26.86 |
| BLEU-3  | 15.46 | 15.46 | 15.46 | 15.46 | 15.28 | 15.46 | 15.46 | 15.07 | 15.46 | 15.26 | 15.60 | 15.46 | 15.46 | 15.46 | 15.46 | 15.49 | 15.46 | 15.46 |
| BLEU-4  | 8.61 | 8.61 | 8.61 | 8.61 | 8.29 | 8.61 | 8.61 | 8.27 | 8.61 | 8.70 | 8.75 | 8.61 | 8.61 | 8.61 | 8.61 | 8.72 | 8.61 | 8.61 |
| ROUGE-R | 32.92 | 32.92 | 32.92 | 32.92 | 32.47 | 32.92 | 32.92 | 32.23 | 32.92 | 32.64 | 33.09 | 32.92 | 32.92 | 32.92 | 32.92 | 33.00 | 32.92 | 32.92 |
| ROUGE-P | 40.82 | 40.82 | 40.82 | 40.82 | 40.65 | 40.82 | 40.82 | 39.15 | 40.82 | 40.76 | 40.98 | 40.82 | 40.82 | 40.82 | 40.82 | 40.87 | 40.82 | 40.82 |
| ROUGE-F | 36.36 | 36.36 | 36.36 | 36.36 | 36.03 | 36.36 | 36.36 | 35.29 | 36.36 | 36.18 | 36.53 | 36.36 | 36.36 | 36.36 | 36.36 | 36.43 | 36.36 | 36.36 |

Table 10: Subject-wise evaluation results on a model trained with subject **YDG**, where the radar chart suggests the performance variance on different subjects on each metric.

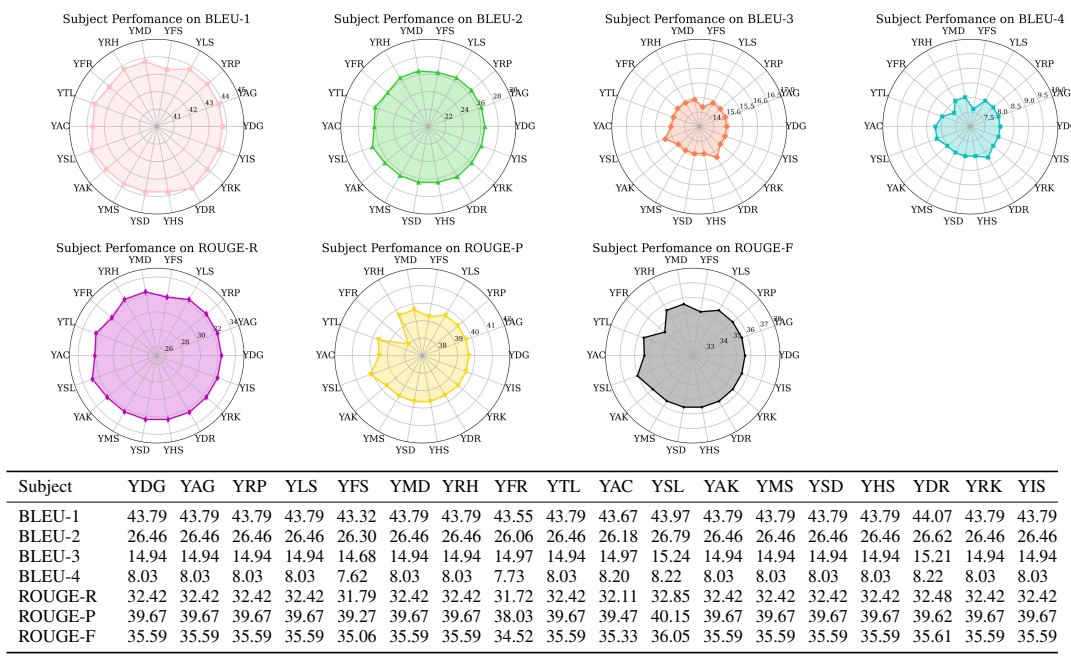

| Subject | YDG | YAG | YRP | YLS | YFS | YMD | YRH | YFR | YTL | YAC | YSL | YAK | YMS | YSD | YHS | YDR | YRK | YIS |
|---------|-----|-----|-----|-----|-----|-----|-----|-----|-----|-----|-----|-----|-----|-----|-----|-----|-----|-----|
| BLEU-1  | 43.79 | 43.79 | 43.79 | 43.79 | 43.32 | 43.79 | 43.79 | 43.55 | 43.79 | 43.67 | 43.97 | 43.79 | 43.79 | 43.79 | 43.79 | 44.07 | 43.79 | 43.79 |
| BLEU-2  | 26.46 | 26.46 | 26.46 | 26.46 | 26.30 | 26.46 | 26.46 | 26.06 | 26.46 | 26.18 | 26.79 | 26.46 | 26.46 | 26.46 | 26.46 | 26.62 | 26.46 | 26.46 |
| BLEU-3  | 14.94 | 14.94 | 14.94 | 14.94 | 14.68 | 14.94 | 14.94 | 14.97 | 14.94 | 14.97 | 15.24 | 14.94 | 14.94 | 14.94 | 14.94 | 15.21 | 14.94 | 14.94 |
| BLEU-4  | 8.03 | 8.03 | 8.03 | 8.03 | 7.62 | 8.03 | 8.03 | 7.73 | 8.03 | 8.20 | 8.22 | 8.03 | 8.03 | 8.03 | 8.03 | 8.22 | 8.03 | 8.03 |
| ROUGE-R | 32.42 | 32.42 | 32.42 | 32.42 | 31.79 | 32.42 | 32.42 | 31.72 | 32.42 | 32.11 | 32.85 | 32.42 | 32.42 | 32.42 | 32.42 | 32.48 | 32.42 | 32.42 |
| ROUGE-P | 39.67 | 39.67 | 39.67 | 39.67 | 39.27 | 39.67 | 39.67 | 38.03 | 39.67 | 39.47 | 40.15 | 39.67 | 39.67 | 39.67 | 39.67 | 39.62 | 39.67 | 39.67 |
| ROUGE-F | 35.59 | 35.59 | 35.59 | 35.59 | 35.06 | 35.59 | 35.59 | 34.52 | 35.59 | 35.33 | 36.05 | 35.59 | 35.59 | 35.59 | 35.59 | 35.61 | 35.59 | 35.59 |

Table 11: Subject-wise evaluation results on a model trained with subject **YFS**, where the radar chart suggests the performance variance on different subjects on each metric.

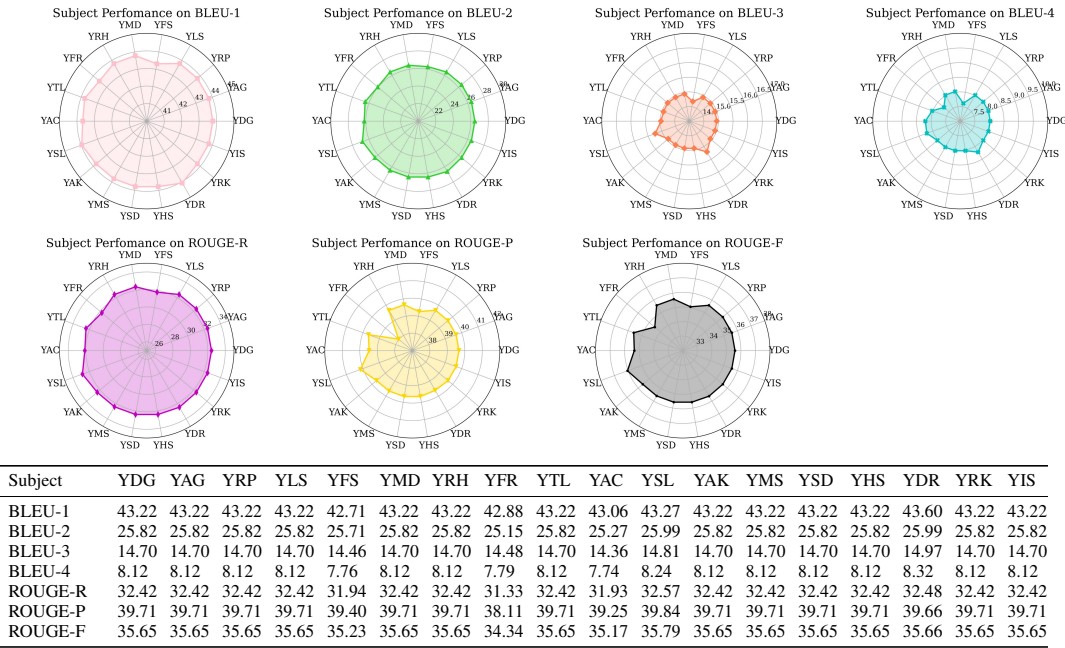

| Subject | YDG | YAG | YRP | YLS | YFS | YMD | YRH | YFR | YTL | YAC | YSL | YAK | YMS | YSD | YHS | YDR | YRK | YIS |
|---|---|---|---|---|---|---|---|---|---|---|---|---|---|---|---|---|---|---|
| BLEU-1 | 43.22 | 43.22 | 43.22 | 43.22 | 42.71 | 43.22 | 43.22 | 42.88 | 43.22 | 43.06 | 43.27 | 43.22 | 43.22 | 43.22 | 43.22 | 43.60 | 43.22 | 43.22 |
| BLEU-2 | 25.82 | 25.82 | 25.82 | 25.82 | 25.71 | 25.82 | 25.82 | 25.15 | 25.82 | 25.27 | 25.99 | 25.82 | 25.82 | 25.82 | 25.82 | 25.99 | 25.82 | 25.82 |
| BLEU-3 | 14.70 | 14.70 | 14.70 | 14.70 | 14.46 | 14.70 | 14.70 | 14.48 | 14.70 | 14.36 | 14.81 | 14.70 | 14.70 | 14.70 | 14.70 | 14.97 | 14.70 | 14.70 |
| BLEU-4 | 8.12 | 8.12 | 8.12 | 8.12 | 7.76 | 8.12 | 8.12 | 7.79 | 8.12 | 7.74 | 8.24 | 8.12 | 8.12 | 8.12 | 8.12 | 8.32 | 8.12 | 8.12 |
| ROUGE-R | 32.42 | 32.42 | 32.42 | 32.42 | 31.94 | 32.42 | 32.42 | 31.33 | 32.42 | 31.93 | 32.57 | 32.42 | 32.42 | 32.42 | 32.42 | 32.48 | 32.42 | 32.42 |
| ROUGE-P | 39.71 | 39.71 | 39.71 | 39.71 | 39.40 | 39.71 | 39.71 | 38.11 | 39.71 | 39.25 | 39.84 | 39.71 | 39.71 | 39.71 | 39.71 | 39.66 | 39.71 | 39.71 |
| ROUGE-F | 35.65 | 35.65 | 35.65 | 35.65 | 35.23 | 35.65 | 35.65 | 34.34 | 35.65 | 35.17 | 35.79 | 35.65 | 35.65 | 35.65 | 35.65 | 35.66 | 35.65 | 35.65 |

Table 12: Subject-wise evaluation results on a model trained with subject **YSL**, where the radar chart suggests the performance variance on different subjects on each metric.

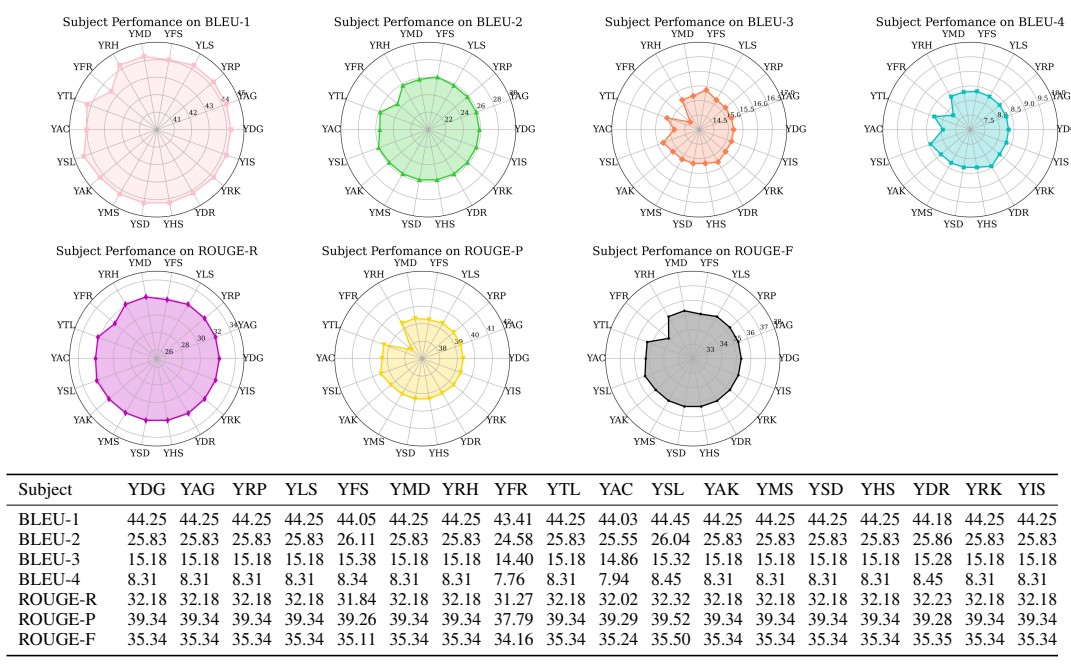

| Subject | YDG | YAG | YRP | YLS | YFS | YMD | YRH | YFR | YTL | YAC | YSL | YAK | YMS | YSD | YHS | YDR | YRK | YIS |
|---|---|---|---|---|---|---|---|---|---|---|---|---|---|---|---|---|---|---|
| BLEU-1 | 44.25 | 44.25 | 44.25 | 44.25 | 44.05 | 44.25 | 44.25 | 43.41 | 44.25 | 44.03 | 44.45 | 44.25 | 44.25 | 44.25 | 44.25 | 44.18 | 44.25 | 44.25 |
| BLEU-2 | 25.83 | 25.83 | 25.83 | 25.83 | 26.11 | 25.83 | 25.83 | 24.58 | 25.83 | 25.55 | 26.04 | 25.83 | 25.83 | 25.83 | 25.86 | 25.83 | 25.83 |
| BLEU-3 | 15.18 | 15.18 | 15.18 | 15.18 | 15.38 | 15.18 | 15.18 | 14.40 | 15.18 | 14.86 | 15.32 | 15.18 | 15.18 | 15.18 | 15.18 | 15.28 | 15.18 | 15.18 |
| BLEU-4 | 8.31 | 8.31 | 8.31 | 8.31 | 8.34 | 8.31 | 8.31 | 7.76 | 8.31 | 7.94 | 8.45 | 8.31 | 8.31 | 8.31 | 8.31 | 8.45 | 8.31 | 8.31 |
| ROUGE-R | 32.18 | 32.18 | 32.18 | 32.18 | 31.84 | 32.18 | 32.18 | 31.27 | 32.18 | 32.02 | 32.32 | 32.18 | 32.18 | 32.18 | 32.18 | 32.23 | 32.18 | 32.18 |
| ROUGE-P | 39.34 | 39.34 | 39.34 | 39.34 | 39.26 | 39.34 | 39.34 | 37.79 | 39.34 | 39.29 | 39.52 | 39.34 | 39.34 | 39.34 | 39.34 | 39.28 | 39.34 | 39.34 |
| ROUGE-F | 35.34 | 35.34 | 35.34 | 35.34 | 35.11 | 35.34 | 35.34 | 34.16 | 35.34 | 35.24 | 35.50 | 35.34 | 35.34 | 35.34 | 35.34 | 35.35 | 35.34 | 35.34 |

Table 13: Subject-wise evaluation results on a model trained with subject **YMD**, where the radar chart suggests the performance variance on different subjects on each metric.