# OpenReview forum: "DeWave: Discrete Encoding of EEG Waves for EEG to Text Translation"
_NeurIPS.cc/2023/Conference — NeurIPS 2023 spotlight_

### Official Review · Reviewer_E1CL · 2023-07-03

**Soundness:** 3 good
**Presentation:** 3 good
**Contribution:** 2 fair
**Rating:** 6
**Confidence:** 3

**Summary:**

This paper proposes an EEG to text model that feeds into the raw EEG signals and predicts the corresponding words or long sentences. The model is optimized in two stages: (i) matching EEG to Text by contrastive learning with codex quantization; (ii) fintune the BART model for EEG embedding decoding to text. The model is evaluated on ZuCo 1.0 and 2.0 datasets.

**Strengths:**

1. The EEG to text application sounds interesting and challenging. The proposed model architecture sounds reasonble by leveraging recent popular CLIP-style contrastive learning, signal quantization (from speech processing domain), and BART finetuning (LLM).
2. The paperr is easy to read, and the related works are well organized.
3. The performance on ZuCo datasets looks good.


**Weaknesses:**

1. The reviewer has concerns on the data volume for training the CLIP model (ZuCo dataset might not be large enough). It would be good if authors can comment/explain on this.
2. It would be good to add more experiments on "whether the discrete codex" module is useful? What if we directly use the raw continuous signal? How to better learn the discreate codex (the codex grouping in [1] can potentially improve the model)?
3. BART is a generally purpose pre-trained language model, which may not be suitable for this application. In the medical domain, [2] might be more suitable given the finetuned data might be limited.
4. More datasets and baseline models should be added to systematically and rigorously demonstrate the supriority of the proposed model.

[1] Baevski et al. Wav2vec 2.0: A Framework for Self-Supervised Learning of Speech Representations.
[2] Lee et al. BioBERT: a pre-trained biomedical language representation model for biomedical text mining


**Questions:**

NA

---

> ### Author Rebuttal · Authors · 2023-08-07
>
> We appreciate the reviewer's thoughtful feedback and recognize the significance of the highlighted concerns. Below, we address the identified weaknesses point by point:
>
> 1. **Data Volume Concerns for the CLIP Model**: The limited data scale problem has been a key challenge in the BCI area. The training data is limited compared to visual-language alignment CLIP addressing. This also leads to EEG-to-Text alignment remains unexplored compared to Vision-to-Text alignment. However, this work proves that introducing contrastive alignment on a limited data scale has a clear improvement and cross-modality alignment in the BCI area is feasible, which can be a good reference/encouragement for future works. Although the text corpus is limited, each sentence has EEG waves from 18/12 human subjects reading, this still leads to a large amount of EEG data. We utilize pre-trained language models as the decoder and pre-trained word2vec embedding as the alignment guidance for the EEG encoder, which alleviates the limited training data problem.
>
> 2. **Utility of the Discrete Codex Module**: This is a good point. For comparison directly using raw continuous signal, we have already reported in raw waves part in Tab. 1 in the main paper. Since the EEG-to-Text [1] is only designed for word-level eeg features. We re-implement this method and directly train it on raw waves to realize this comparison. For a fair comparison, both methods are using the same pre-trained BART decoder. We list the performance reported in Table 1 in the main paper (between line 213 to line 243) here for your reference.
>
>    | Method                              | BLEU-1 | BLEU-2 | BLEU-3 | BLEU-4 | ROUGE-R | ROUGE-P | ROUGE-F |
>    | ----------------------------------- | ------ | ------ | ------ | ------ | ------- | ------- | ------- |
>    | EEG-to-Text (directly on raw waves) | 13.07  | 5.78   | 2.55   | 1.10   | 15.22   | 18.08   | 16.36   |
>    | DeWave                              | 20.51  | 10.18  | 5.16   | 2.52   | 21.18   | 29.42   | 24.27   |
>
>    The performance suggests a clear improvement in using discrete codex over directly using continuous raw EEG waves.
>
>    Regarding the open question of grouping the discrete codex, we think given the current limited training data further splitting the codex into sub-groups might not have a positive effect. But it could be an interesting question in the future if there is sufficient data. It is also noted that, according to Fig. 5 of the main paper, simply increasing codex size will lead to a decrease in performance. In that case, we didn’t report these aspects in the main paper.
>
> 3. **Choice of BART over Domain-specific Models**:  The choice of BART has three reasons: 1) as you mentioned, it is a general-purpose language model with normal language distributions. As ZuCo uses text corpus mostly from wiki pages in the general domain, not the medical domain, the pre-trained distribution of BART is closer to the ZuCo dataset. 2) BART pretraining process added noise tokens to the encoder when training the language decoder. This is more suitable for noisy encoded EEG embedding. 3) In order to control the same decoder model with previous works [1]. In that case, we could better illustrate the impact on the proposed discrete codex with contrastive alignment.
>
> 4. **Possibility of Additional Datasets and Baselines**:  We very much like to have more data on brain signal-to-text translation. However, subject to the current situation as of the submission deadline, the ZuCo dataset is the only choice. We will actively track the new datasets. Also, we are recording our own data currently and will provide more details in our future work.
>
>    For baselines, the EEG-to-Text [1] baseline in our main paper is the only valid baseline currently. We've tried our best to adopt possible baselines from other domains for this task. We adopt Wav2Vec [2] from the speech recognition area and SCL from the brain sleeping stage recognition area and BENDR [3] from EEG self-supervised training area into translation tasks as our baselines. Yet, in this case, we think the current experiment is quite completely limited to existing works.
>
> Thank the reviewer for the very good advice, we will actively track the baselines of brain decoding for the community in our future works.
>
> Again, we thank the reviewer for their constructive feedback and hope our rebuttal addresses your concern.
>
> [1] Zhenghailong Wang, et al, **Open Vocabulary Electroencephalography-to-Text Decoding and Zero-Shot Sentiment Classification** AAAI 2022.
>
> [2] Baevski, Alexei, et al. **Wav2vec 2.0: A Framework for Self-Supervised Learning of Speech Representations.** NeurIPS 2020: 12449-12460.
>
> [3] Kostas, Demetres, et al. **BENDR: Using Transformers and a Contrastive Self-Supervised Learning Task to Learn From Massive Amounts of EEG Data.** Frontiers in Human Neuroscience 15 (2021).

---

> > ### Comment · Reviewer_E1CL · 2023-08-18
> > **Thanks for your rebuttal.**
> >
> > We thank the authors for detailed rebuttal and additional experiments. I still think the small dataset might not be sufficient to give solid insights and conclusions on the BCI problem. I will keep my score.

---

### Official Review · Reviewer_iEZr · 2023-07-04

**Soundness:** 3 good
**Presentation:** 2 fair
**Contribution:** 3 good
**Rating:** 7
**Confidence:** 3

**Summary:**

This paper introduces a new method, “DeWave”, for decoding text strings from EEG data recorded from subjects while reading. This method seems to differ from earlier ones in two important and useful ways: (1) it uses a discrete “codebook” to represent the EEG data, which helps to control noise in the extremely noisy setting of EEG, and (2) it is also applied to continuous, rather than word-segmented, EEG data, more closely mimicking a real-world situation. The quantitative results show a modest boost over earlier methods in the word-aligned setting, and decent performance in the novel continuous setting.

**Strengths:**

* This paper is the first I have seen that attempts to decode continuous language, i.e. without explicit word boundary markers, from EEG data. This is a big and important step! Decoding from word-aligned EEG snippets is clearly much easier but also much less interesting than this variant. I think pushing for this type of decoding is a great contribution of the current work.
* The use of a vector quantization approach to represent EEG data also seems like an important step. (However, I have not read every paper in this area, so I can’t say with 100% confidence that this is the first work to apply this approach to the current problem.)
* The method seems, overall, well-designed.

**Weaknesses:**

* The paper is at several points confusing and difficult to follow or understand. For example, in section 3.2 I found it impossible to understand the sentence “The codex contains fewer time-wise properties which could alleviate the order mismatch between event markers (eye fixations) and language outputs.” (I’ll expand on this with specific queries under “Questions”, below.)
* Minor scientific issue: Eye movements were used to segment the EEG data into words in the segmented (traditional) condition, but were ignored for the continuous condition. However, eye movements are well known to cause HUGE transient deflections in EEG signals, to the point where much/most EEG data includes preprocessing that attempts to minimize their influence. I would imagine such preprocessing was done for this dataset (please correct me if I’m wrong), but even ideal eye movement removal cannot remove 100% of the related signals. So I think it could be important for the authors to consider (and perhaps even test) whether their continuous decoder is actually discovering latent eye movement signals.

**Questions:**

* In section 3.2 under “Inference”, it would be very useful to state how temporal information is encoded, e.g. with a separate x for each word in the segmented condition, and for each 200 ms time period in the continuous condition.
* In section 3.3 under “Word-Level EEG Features…”, the preprocessing is not well explained. This section might rely on the reader knowing earlier work, but I found it difficult to understand what the 840-dimensional vector actually comprises.
* I found it very surprising and slightly worrying that the decoder seemed to be able to retrieve both exact proper nouns (“Kerouac” in Table 3, example 3) and approximate proper nouns (“Heroughs” for “Burroughs” in same). How is this possible? The contrastive semantic alignment loss, as I understand, cannot account for this, since it would operate on discrete tokens, unless those tokens also occurred in the training set. Is the discrete codex really capturing enough information about the constituent BPE tokens for those words to reconstruct them?

**Limitations:**

I think the most important limitation to note is that this method applies to EEG data collected while subjects actively read text, and cannot (currently) be applied to data collected while subjects merely think words. The authors do note this in their “Limitations” section, which is great.

---

> ### Author Rebuttal · Authors · 2023-08-07
>
> Dear reviewer,
>
> Thank you for your thorough review and constructive comments on our manuscript. We genuinely appreciate your insights and will address your concerns point by point below.
>
> 1. **In section 3.2 under “Inference”, it would be very useful to state how temporal information is encoded.**
>
>    - For the word level EEG features, since there are eye-tracking markers suggesting EEG wave fragments related to each word, we directly get the embedding sequence by slicing it into sequence {$x_p$},  $p$={$1,2,..M$}, where $p$ th embedding corresponding to each word.
>    - For the raw wave setting, as shown in Fig. 3, we use a wide convolutional kernel to slide through a certain stride length to sequentially slice raw waves into continuous embedding sequences {$x_p$},  $p$={$1,2,..N$}, where $p$ th embedding corresponding to continuous perception fields (take 200ms as an example).
>
>    Then we directly use the position embedding in the original transformer paper [1] for both settings. Given position (order $p$), the position embedding in dimension $i$ is calculated by $PE_{(p,2i)}=sin(p/10000^{2i/d_{model}}), \quad PE_{(p,2i+1)}=cos(p/10000^{2i/d_{model}})$.
>
>    As the calculated position embedding is with the same shape as the continuous embedding sequence $\{x_p\}$, the position embedding is directly added with each EEG embedding accordingly. The position embedding is intrinsically maintained in the sequence.
>
>    **Further question: how does codex benefit the feature noise due to time-wise properties**
>
>    When a human subject reads the same word, the EEG waves could be different, given in different sentences context, in a different order, and at different times, which leads to feature variance time-wise. The advances of discrete encoding are that it could minimize noise variances and intuitively map encoded semantic (might with noise) embedding corresponding to the same word into a stable discrete codex value representation for the decoder. This will benefit the decoder with a more stable feature representation with less time-wise variances.
>
>    We will significantly improve writing in Section 3.2 for better readability.
>
> 2. **Eye Movements and EEG Data:** Yes, you are right in stating pre-processing procedures the dataset used, preprocessing steps were indeed taken to minimize the influence of eye movements. Here [2], 9 EOG channels were used for artifact removal, and additional 14 channels lying mainly on the neck and face were discarded before data analysis. Fig. 1d in paper [2] shows a very good removal performance.
>
>    However, there may remain a small noisy component related to eye tracking as the artifact removal is not 100% accurate. Yet, according to both our correlation analysis experiments and experiments reported in EEG-Text [3]’s code, we have not observed a significant component related to eye movement.
>
>    Another thing is we use a quite deep transformer encoder to extract signals to semantic space, even if it has some remained eye-movement information, it will be jointly encoded into semantic embeddings. In that case, it could be hard to separate the impact from eye movement individually. However, we are keen to explore more properties of this point and keep updating it.
>
> 3. **section 3.3 under “Word-Level EEG Features...” and how the feature dimension 840 comes from:** Thank you very much for the writing suggestion to improve readability to wider audiences who are not familiar with earlier work and the dataset. The EEG waves are collected from the Biosemi-128 system, where 9 EOG channels were used for artifact removal, and additional 14 channels lying mainly on the neck and face were discarded. That led to 105 channels for EEG signals. For word-level features, the eye-tracking marker is used to slice EEG wave fragments according to each word. Then, both [2, 3] and our paper averaged the power statistical feature on 8 frequency bands. These bands are 'theta\_{1,2} (4-6 Hz, 6.5-8 Hz)', 'alpha\_{1,2} (8.5-10 Hz, 10.5-13 Hz)', 'beta\_{1,2} (13.5-18 Hz, 18.5-30 Hz)', 'gamma_{1,2} (30.5-40 Hz, 40-49.5 Hz)'. In that case, the feature shape after flattening and concatenating all statistical features would be $8\times 105=840$.
>
>    We will enhance this section by providing a step-by-step description of our preprocessing pipeline, ensuring it is self-contained and does not necessitate prior knowledge of earlier works [2, 3].
>
> 4. **Decoded example text:**
>
>    According to our experiments, the decoder has better prediction ability on nouns rather than verbs and adjectives. Our observation is that when human subject reading, it normally pays more attention to nouns (names) (which are likely to be more complicated, or even rarely seen) rather than common words. It leads to enriched input feature qualities for the model to learn.
>
>    Another point is the data is recorded when the human subject is reading the text. Considering the mentioned term “Burroughs and Kerouac” vs. “Heroughs and Kerouac”, it has similar pronunciation when reading on “-roughs” part. In that case, the learned distribution might be closer between these words (adjusted from word2vec).
>
>    Meanwhile, according to our statistical results on the ZuCo dataset, it contains 6828 total words and 300 sentences (1.0) and 390 sentences (2.0). Though the training and testing text corpus is different, these corpora are sampled from the same wiki paragraph. So it is true that these two names did appear in the training set. The current sampled example text is surprising, but we think this performance is totally under control and as we expected.
>
>
>
> [1] Vaswani, Ashi et al, **Attention is all you need** , NeurIPS 2017
>
> [2] Nora Hollenstein, et al, **ZuCo 2.0: A Dataset of Physiological Recordings During Natural Reading and Annotation.** LREC 2020: 138-146
>
> [3] Zhenghailong Wang, et al, **Open Vocabulary Electroencephalography-to-Text Decoding and Zero-Shot Sentiment Classification** AAAI 2022

---

> > ### Comment · Reviewer_iEZr · 2023-08-17
> >
> > Thank you for these updates and clarifications. I did not realize that the stimuli in the ZuCo datasets consisted of wikipedia excerpts. Does it worry you that the LLMs you used (BART, OPT, & Llama) all (almost certainly) included those wikipedia pages in their training sets?

---

> > > ### Author Response · Authors · 2023-08-18
> > >
> > > Thank you to the reviewer for the follow-up questions.
> > >
> > > The authors do agree that we should be cautious when dealing with LLMs when bridging them with brain waves. The experimental results fit our expectations and are safe in ethics. The authors do not worry much about the point that LLMs included Wikipedia pages in their training set considering the following aspects.
> > >
> > > 1. The **primary objective** of this paper is to **develop a better brain encoding** that aligns more effectively with language model decoders. In our main experiments, we maintained the decoder consistent with baseline models, focusing our comparison on the quality of the learned encoding. With the same decoder, the introduced encoding surpasses previous top results by clear margins, particularly in the raw wave setting. This supports our contribution.
> > >
> > > 2. The experimental setting involves an open-vocabulary brain-to-text translation task. For open-vocabulary tasks, such as Visual Question Answering (VQA) or image captioning, a certain degree of text phase overlap between the training and test sets is permissible, especially as the training set size increases. Given the vastness of the training corpus for current Large Language Models (LLMs), most people agree that these modern auto-regressive models are learning a joint distribution that inherently encompasses 'concepts' rather than merely replicating the exact same phrases.
> > >
> > >    Additionally, there are 18 different human subjects reading the same text sample, which already creates a lot of **diversity**  in the input feature. Directly decoding language from brain waves is a quite challenging task. If the encoded embeddings align well with the pre-trained language decoder, it then signifies a reduction in the modality gap between the two modalities, which could benefit follow-up works in the EEG encoding area.
> > >
> > > 3. The qualitative results also support this point. As evidenced by Table 3, the generated examples are not merely regurgitating Wikipedia excerpts. Rather, they appear to reconstruct the stimuli derived from human subjects during reading, even though the resulting sentences may lack fluency in their semantic coherence. If the decoder is merely reproducing previously encountered content, the text will exhibit greater fluency but diminished correlation.

---

> > > > ### Comment · Reviewer_iEZr · 2023-08-18
> > > >
> > > > Thank you for these clarifications. I agree on all points. I am raising my score to a 7.

---

### Official Review · Reviewer_tXBq · 2023-07-06

**Soundness:** 3 good
**Presentation:** 3 good
**Contribution:** 2 fair
**Rating:** 6
**Confidence:** 4

**Summary:**

The authors propose a new framework called DeWave, which integrates discrete encoding sequences with EEG-to-text translation tasks, using a quantized variational encoder and pre-trained language models. This approach overcomes the mismatch between eye fixations and spoken words and reduces interference from individual differences in EEG waves. The model outperforms previous baselines on the ZuCo Dataset, achieving improved BLEU-1 and Rouge-F scores. Importantly, this work is the first to enable translation of entire EEG signal periods without relying on word-level order markers like eye fixations.

**Strengths:**

- The authors introduce a novel discrete codex encoding to EEG waves, which seems like a promising way of representing EEG signals.
- DeWave has promising results as it achieves state of the art performances on EEG to text translation.

**Weaknesses:**

- Figure 5's caption reads "Ablation study on different codex sizes and perception fields (raw waves)," however, for the table on the left, the experiments are done on word level EEG features. It would be better if the table title on perception field can specify the feature type (raw waves) instead of the caption.
- In the abstract, the authors mention large language models such as ChatGPT, but experiment only on BART. An ablation study on other LLMs such as LLaMA, BERT, or RoBERTa can be enlightening.
‌

**Questions:**

- The authors seem to have conducted the experiments separately for each patient or an averaged score over all patients. In [1], they mention "In spite of the high inter-subject variability in EEG data, it has been shown in previous research of machine learning applications (Foster et al., 2018; Hollenstein et al., 2019a), that averaging over the EEG features of all subjects yields results almost as good as the single best-performing subjects." Have the authors considered this method?
- I am aware that the ZuCo dataset has sentence level features. Additionally, I am aware of some works that have experimented on concatenated word level features [2]. Have the authors considered these different EEG feature types?


[1] Hollenstein N, Renggli C, Glaus B, Barrett M, Troendle M, Langer N, Zhang C. Decoding EEG Brain Activity for Multi-Modal Natural Language Processing. Front Hum Neurosci. 2021 Jul 13;15:659410. doi: 10.3389/fnhum.2021.659410. PMID: 34326723; PMCID: PMC8314009.
[2] Han, William, et al. “An Empirical Exploration of Cross-Domain Alignment between Language and Electroencephalogram.” ArXiv:2208.06348 [Cs, Q-Bio], 10 Aug. 2022, arxiv.org/abs/2208.06348.

**Limitations:**

The authors have adequately addressed the limitations.

---

> ### Author Rebuttal · Authors · 2023-08-07
>
> Dear Reviewer,
>
> We appreciate your thorough review and insightful feedback. We will address each of your comments and concerns below and also in our revised manuscript.
>
> 1. **Improve Figure 5’s title**: Thank you for your suggestion on slight confusion on the perception field graph. We will revise the figure title to ``Ablation Metrics on Perception Fields (on Raw Waves)” which ensures that the table title specifies the feature type (raw waves) rather than the current caption for clarity. This should provide better readability to the audience with less background knowledge.
>
> 2. **Experimentation on other LLMs:** We agree with the reviewer’s suggestion that further extending the discrete codex for large language models can be enlightening. The reason why we keep the main experiments using BART is that we want to control the decoder on the same scale compared to previous works. In that case, we can better illustrate that our improvement is from discrete coding but not from a more powerful decoder.
>
>    In fact, we did have applied a larger-scale ablation in the near past. We conducted further experiments by replacing the BART decoder with OPT and Llama V1. However, the performance improvement is not as large as we expected. Bridging brain activities with LLMs and AGI is an important area worth plenty of papers to explore. **For reasons of caution, we did not include** this experiment previously in the main paper.
>
>    However, we can report our previous experiments on these three models. Limited to our computing resources, we fine-tune the OPT-1.3B model and Llama-1 7B model with half-precision each for 3 epochs using PyTorch FSDP mode. The tokenized EEG waves are prompted into LLMs according to Mini-GPT4’s method [1] of dealing visual tokens.
>
>    | Source              | Decoder             | BLEU-1 | BLEU-3 | ROUGE-R | ROUGE-P | ROUGE-F |
>    | ------------------- | ------------------- | ------ | ------ | ------- | ------- | ------- |
>    | Word-level features | DeWave              | 41.35  | 13.92  | 28.82   | 33.71   | 30.69   |
>    | Word-level features | DeWave + OPT 1.3B   | 41.97  | 14.06  | 28.98   | 33.82   | 30.86   |
>    | Word-level features | DeWave + Llama-1 7B | 42.84  | 15.03  | 29.42   | 35.43   | 32.05   |
>    |                     |                     |        |        |         |         |         |
>    | Raw Waves           | DeWave              | 20.51  | 5.16   | 21.18   | 29.42   | 24.27   |
>    | Raw Waves           | DeWave + OPT 1.3B   | 21.31  | 5.84   | 22.09   | 29.94   | 25.42   |
>    | Raw Waves           | DeWave + Llama-1 7B | 22.05  | 6.03   | 22.45   | 30.01   | 26.08   |
>
> 3. **Averaging over the EEG features:** Considering potential practical use cases in the future, using averaged EEG waves might not be very valuable. This is because, in real-world scenarios, the translation from brainwaves to text is more likely to be performed on individual human subjects in real time. If our understanding is right, averaging waves across multiple subjects requires pre-collecting EEG waves offline and performing benchmark tests. Given intuition closer to realistic usage, we trained on a mixed dataset and tested on different single human subjects to better reflect real-world applicability.
>
> 4. **Different EEG feature types:** Yes, we considered this method previously. There are two reasons why we didn't use it.
>
>      - As the EEG-to-Text translation is conducted on long sentences, which is different from traditional classification tasks. The EEG feature fragments according to each word have large variation according to how many times the word appears in the sentence and how many times human subjects looks at that word when reading. In that case, concatenate features lead to higher computational consumption and instability in input features.
>
>
>      - We have drafted experiments in the earlier stage. According to our previous experiments, the performance of concatenate method is significantly lower than averaging word-level feature, which is used by [2] and our paper.
>
>
>
>
> [1] Deyao Zhu et al,  **MiniGPT-4: Enhancing Vision-language Understanding with Advanced Large Language Models**
>
> [2] Zhenghailong Wang, et al, **Open Vocabulary Electroencephalography-to-Text Decoding and Zero-Shot Sentiment Classification** AAAI 2022

---

> > ### Comment · Reviewer_tXBq · 2023-08-20
> >
> > Thanks for the insightful rebuttal and clarification to my questions! I will raise my score to a 6. I wish the authors the best of luck.

---

### Official Review · Reviewer_TbVs · 2023-07-06

**Soundness:** 3 good
**Presentation:** 2 fair
**Contribution:** 3 good
**Rating:** 7
**Confidence:** 4

**Summary:**

The authors present an approach to decode language from EEG data. The proposed approach can work on both time-locked and raw data (i.e. without markers indicating when a word was read). The model is trained to (1) reconstruct its EEG input using a vector-quantized representation (learnable "codex") in a pretraining stage, (2) align the learned codex representations with word2vec embeddings of the corresponding words and (3) fine-tune the whole model end-to-end including a language model. Experiments on a text-EEG dataset are presented, showing the proposed approach outperforms existing baselines including other self-supervised learning objectives that do not use vector quantization on BLEU and ROUGE metrics. Ablation studies on codex size and windowing parameters are presented, along with  an analysis of cross-subject performance. Finally, examples of EEG-to-text decoding are also presented.

**Strengths:**

Originality: The proposed approach combining vector quantized representation learning and "freeform" text generation from decoded EEG is novel.

Quality: The submission appears technically sound, with claims supported by benchmark comparisons and ablation studies.

Clarity: The paper is mostly clear, though some details are hard to understand from the text (see questions).

Significance: I believe these results pave the way to better text decoding from EEG data. This is important for the field of brain decoding given a lot of the work on brain-to-text has been using intracranial modalities or fMRI, which are very expensive and dramatically more constraining than surface EEG.

**Weaknesses:**

The main weakness of this submission in my opinion is that several elements remained unclear after reading the text and taking a look at the provided code. For instance, I did not understand how the alignment was achieved between raw EEG and words/text (Question 2), how the language model was fine-tuned (Question 3) and how DeWave was combined with existing SSL approached (Question 4).

**Questions:**

1. Section 3.3, lines 136-141: The description in this paragraph is unclear to me. What is meant by "statistical result of four frequency bands"? I assume that summary statistics of the power in the four different bands were used? Also, the phrase "different fragments may have different wavelengths" is confusing as I believe the same frequency bands are used for each EEG fragment. More generally, why is a "handcrafted feature" approach used for the word-level features, while a similar convolutional encoder as is used for the "raw wave" could instead be trained end-to-end? The ablation of Table 4 shows that pretraining the word-level models improves performance bit a small amount, and I wonder if performance would further improve with a learned tokenization.
2. I do not understand how the sequence of embeddings obtained from the raw EEG were aligned with word2vec representations (Section 3.3, second paragraph). In the word-level experiments, my understanding is that each word was embedded with word2vec, and that the resulting embedding was used to align the EEG-based discrete representations with the loss of Eq.4. However in the "raw" case, how was word2vec applied to the text, i.e. was it still applied at the word level?
3. What loss is used to fine-tune the whole system end-to-end including the language model?
4. Table 1 shows that combining DeWave with SCL further improves performance over the base DeWave model. How were these two approaches combined, and did the SCL approach had to be modified?
5. The approach of [2] should be written "wav2vec" instead of "wave2vec" in the text and in Figure 2, Table 1, etc.
6. There is missing information about the dataset (number of subjects, length of data collection, description of the text corpus, etc.).

**Limitations:**

Yes.

---

> ### Author Rebuttal · Authors · 2023-08-07
>
> Thank you very much for your comprehensive review, appreciation, and insightful inquiries. We will address your concern step by step below and improve in the final version.
>
> 1. **Regarding Section 3.3, lines 136-141 and beyond:** This paragraph is to introduce how we construct the word-level EEG features. The method is the same as it is in the ZuCo dataset [1] and EEG-to-Text [2] for fair comparison.
>    - "statistical result of four frequency bands": When the ZuCo dataset preprocesses word-level features, it averages power statistical features on all corresponding EEG fragments of four main bands (two sub-bands for each main band), including 'theta\_{1,2} (4-6 Hz, 6.5-8 Hz)', 'alpha\_{1,2} (8.5-10 Hz, 10.5-13 Hz)', 'beta\_{1,2} (13.5-18 Hz, 18.5-30 Hz)', 'gamma_{1,2} (30.5-40 Hz, 40-49.5 Hz)'.
>    - By "different fragments may have different wavelengths", we meant to emphasize the sliced EEG fragments have **varying time lengths of waves**, not wavelengths. The varying time lengths of EEG waves caused by the eye-tracking markers length variation arises because human subjects spend different amounts of time reading different words. Apologize for the typo here, we will correct “wavelengths” to the “different lengths of EEG wave samples” throughout the whole paper.
>    - Comparing word-level handcraft features with raw waves. Though both are hard tasks, translation on word-level features is easier as :
>      - It already has an EEG feature for word alignment suggested by eye-tracking markers. Instead, the raw wave setting requires the transformer to learn the alignment through self-supervised pre-training as illustrated in Fig.3.
>      - Less noise: Each word-level feature is consistent in each sentence as it averaging feature regardless of how many times the same word appears in the sentence. The performance gap in Table 1 supports this point.
>
>     - Learned tokenization: Thank you for a good point. We think the projection layer is taking a similar role, where it `tokenizes’ handcraft word-level features into transformer space. Tab. 4 suggests that learned tokenization has improvements in the word-level setting.
>
> 2. **Alignment on raw waves:** For the raw waves setting, there are no eye-tracking markers to slice EEG waves accordingly. As shown in Fig. 3, we use a wide convolutional kernel to slide on waves to sequentially slice raw waves time-wise. Then, position embeddings are added into each sliced embedding to maintain the order information. The alignment happens extrinsically sequence-wise (sentence-wise). We simultaneously optimize both the discrete coding and alignment similar to CLIP as described in Eq.4, where alignment matrix $s_{i,j}$ which contains all possible alignment is calculated between each encoded EEG embedding and text embedding in a sequence pair. The transformer attention encoders are expected to learn the alignment $s_{i,j}$ intrinsically given position embedding and output supervision. The current code only contains E2E training, discrete codex, and evaluation, code for contrastive alignment pretraining will be available after the anonymous era.
>
> 3. **End-to-end fine-tuning:** As described between lines 128-132, we maximize the log-likelihood on language output similar to most of the language decoder end-to-end training as $L=-log(P(W |zq (X ))$ for both word-level EEG features and raw waves. The only difference to Eq. 2 is that the discrete codex is fixed during fine-tuning. Thank you very much for the notice, we will make these points more clear for better readability.
>
> 4. **How were DeWave and SCL combined?** Here the +SCL meant we apply the additional contrastive loss proposed in their paper and combine it with a coefficient when training the transformer encoder before discrete codex. As SCL are RNN-based model, we didn’t use their model structure. However, this point is not related to our core contribution. We will clarify this part in our final version and provide a section to introduce details in supplementary details.
>
> 5. **Typo Correction:** Thank you for pointing out the problem regarding "wav2vec". We will correct "wave2vec" to "wav2vec" throughout the text, Fig. 2, Tab. 1, and other potential problems.
>
> 6. **More information about the Dataset:** We use the ZuCo (including both 1.0, 2.0) dataset [1] for our main experiments which is exactly the same as [2]. ZuCo stands for Zurich Cognitive Language Processing Corpus (ZuCo), including raw and preprocessed eye-tracking and electroencephalography (EEG) data. The data is collected by letting human subjects read the given text corpus and simultaneously recording both recording the eye-tracking signal and EEG waves. The collection device is the Biosemi-128 system. After denoising, it provides 105 of 128 channels for down-streaming tasks. The 1.0 data is collected from 12 subjects while the 2.0 data is collected from 18 subjects.
>
>    Regarding the text corpus of the ZuCo dataset, these were sourced from a diverse set of textual genres to ensure a wide variety of syntactic structures and word frequencies, which includes: 1) Wikipedia articles 2) movie reviews 3) BNC (British National Corpus).
>
>    We further apply a statistical analysis on the sentences we utilized from the dataset, and reported below.
>
>    | Feature      | ZuCo 1.0 Natural Reading | ZuCo 2.0 Natural Reading |
>    | ------------ | :----------------------: | :----------------------: |
>    | sentences    |           300            |           390            |
>    | sent. length |       21.3 (±10.6)       |       19.6 (±8.8)        |
>    | total words  |           6386           |           6828           |
>    | word length  |           6.7            |           4.9            |
>
> [1] Nora Hollenstein, et al, **ZuCo 2.0: A Dataset of Physiological Recordings During Natural Reading and Annotation.** LREC 2020
>
> [2] Zhenghailong Wang, et al, **Open Vocabulary Electroencephalography-to-Text Decoding and Zero-Shot Sentiment Classification** AAAI 2022

---

> > ### Comment · Reviewer_TbVs · 2023-08-15
> >
> > Thank you to the authors for their answers to my questions. Some short follow-ups:
> >
> > Q1: Got it, thank you for clarifying this. May I suggest to replace "different lengths of EEG wave samples" by something like "different EEG window sizes"? I believe "window" is clearer than "wave samples" in this context.
> >
> > Q2: In an EEG-text pair $(i,j)$, what is $j$ and how is it matched to $i$? My current understanding is that $j$ is a single word. In that case, and knowing that there is no eye tracking information, how is a "corresponding pair" defined? Couldn't $j$ be (correctly) matched to multiple EEG embeddings?

---

> > > ### Author Response · Authors · 2023-08-16
> > >
> > > Thank you to the reviewer for the questions.
> > >
> > > **A1:** Thank you very much for the suggestion. We agree with the point of changing "different lengths of EEG wave samples" to "different EEG window sizes" will make it clearer. We will revise this phase accordingly throughout the paper.
> > >
> > > **A2:** Yes, your understanding is right. Here the term $i$ denotes $i$-th EEG embedding in the EEG embedding sequence with length $N$. The term $j$ denotes $j$-th word2vec embedding in word sequence with length $N$. The EEG embedding sequences are extracted from EEG raw waves without eye-tracking information and in chronological order by 1) going through conv kernels sliding (transfer continuous raw waves into the raw EEG embedding sequence), 2) transformer encoder (encode the raw sequence into the organized sequence), 3) discrete codex (acquire a compared stable embedding value).
> > >
> > > We calculate the similarity matrix with shape $N\times N$ between these two sequences, each with length $N$. The $s_{i,j}$  denotes the similarity between $i$-th EEG embedding and $j$-th word embedding. Here, the "corresponding pair" $(\hat{i},\hat{j})$ denotes the **diagonal element of that matrix**, where we have $\hat{i}=\hat{j}$.  The contrastive object is to maximize the similarity of the "corresponding pair" and push away others. This means we expect the transformer encoder to learn an organized embedding sequence (compared to a raw sequence) given position embeddings and intrinsic correlations inside the raw sequence. And we expect the learned EEG sequence aligned with the text sequence order.
> > >
> > > This setting is based on the **prior** that the human subject is reading in **chronological order**. According to our observation from both ZuCo data and our data collection experiments, when human subjects are reading/silent speeching, the fixation order is mostly in time-wise order. Thus, we trust the transformer encoder to organize the raw sequence and achieve pre-order alignment (to diagonal).
> > >
> > > The mentioned question of $j$-th word embedding may potentially be matching to multiple embeddings in the EEG sequence: If we understand your question right, this situation exists. Considering a text token sequence ["the", "apple", "is", "on", "the", "table"], there will be two "the" respectively in $0$-th and $4$-th positions. However, according to our analysis of the ZuCo dataset, these multiple matches are mostly meaningless articles like "a", and "the", or verbs such as "is" and "are" in abundance. Considering the fact that when people read the sentences, these articles are merely paid attention to, with very short fixation time, we treat these small proportions of multiple matches as noise. In our implementation, we mask out the duplicated word locations ($s_{0,4}, s_{4,0}$ for this example) outside diagonal locations to prevent conflicts. According to experimental results, this method works well when training on large corpora on the ZuCo dataset and first realizing freeform translation on raw waves.
> > >
> > > We will update more details on the incoming code release, and also make this point clearer in writing accordingly in the paper and supplementary materials.

---

> > > > ### Comment · Reviewer_TbVs · 2023-08-21
> > > >
> > > > Thank you to the authors for the clarifications!

---

### Author Rebuttal · Authors · 2023-08-09

Dear chairs and reviewers,

We express our profound gratitude for the comprehensive feedback and comments on our manuscript. This paper receives **Accept, Weakly Accept, Borderline Accept, and Weakly Accept** during the review period. We are excited about the consensus among the reviewers regarding the novelty and potential impact of our work.

This paper introduces vector quantized representation learning and contrastive alignment between EEG waves and natural languages. The experiments of DeWave showcased state-of-the-art performances on word-level EEG-to-text translation. Additionally, DeWave stands as the pioneering effort to achieve language decoding directly from raw waves. This advancement is a significant step towards real-world applications, which alleviate the dependence on pre-known eye-tracking markers to segment EEG waves accordingly. Meanwhile, the discrete codex encoding introduced for EEG waves also provided a new choice for follow-up works that require EEG wave vectorization.

Building upon the above, we have acted on the reviewers' feedback to enhance the clarity and thoroughness of our paper. Please refer to the rebuttal for each specific review for step-by-step clarification on potential unclear technology inquiries below.  Additionally, we have undertaken a thorough refinement of the manuscript to rectify typos and augment its readability for a broader audience. We are committed to ensuring the manuscript's excellence and its readiness for publication.

The related code provided in the review area will be refined and be public after the anonymous phase.

To conclude, we earnestly believe that our approach can make a good contribution to the EEG decoding realm. The feedback has encouraged us to perfect our manuscript to ensure it can contribute to the community as a solid publication.


Best wishes,

Paper Authors

---

### Decision · Program_Chairs · 2023-09-21

**Decision:**

Accept (spotlight)

**Comment:**

This paper present an ambitious BCI project that is innovative by (1) using EEG, a much more noisy yet much more feasible tool for everyday BCI than what is currently used in papers (e.g., ecog, fmri) and (2) by decoding connected input (read text) through the use of eye movements instead of doing the easier task of cutting the EEG signal into words prior to decoding. These steps make the current work very useful for future BCI technology. The reviewers have a good assessment of the paper after the responses, and the authors are requested to include the required clarifications and new experiments in their manuscript or supplementary.